# Solving Continuous Mean Field Games: Deep Reinforcement Learning for Non-Stationary Dynamics

**Lorenzo Magnino**[*]
University of Cambridge

**Kai Shao**[†]
KTH Royal Institute of Technology

**Zida Wu**
University of California, Los Angeles

**Jiacheng Shen**
NYU Center for Data Science

**Mathieu Laurière**
NYU Shanghai

## Abstract

Mean field games (MFGs) have emerged as a powerful framework for modeling interactions in large-scale multi-agent systems. Despite recent advancements in reinforcement learning (RL) for MFGs, existing methods are typically limited to finite spaces or stationary models, hindering their applicability to real-world problems. This paper introduces a novel deep reinforcement learning (DRL) algorithm specifically designed for non-stationary continuous MFGs. The proposed approach builds upon a Fictitious Play (FP) methodology, leveraging DRL for best-response computation and supervised learning for average policy representation. Furthermore, it learns a representation of the time-dependent population distribution using a Conditional Normalizing Flow. To validate the effectiveness of our method, we evaluate it on three different examples of increasing complexity. By addressing critical limitations in scalability and density approximation, this work represents a significant advancement in applying DRL techniques to complex MFG problems, bringing the field closer to real-world multi-agent systems.

## 1 Introduction

Learning in multiplayer games poses significant challenges due to the interplay between strategic decision-making and the dynamics, often non-stationary, interactions among agents. Deep reinforcement learning (DRL) has recently achieved remarkable success in two-player or small-team games such as Go [Silver et al., 2016, 2018], chess [Silver et al., 2017], poker [Heinrich and Silver, 2016], StarCraft [Samvelyan et al., 2019], and Stratego [Perolat et al., 2022]. However, scaling these methods to large populations of agents remains difficult. As the number of agents grows, the joint strategy space becomes prohibitively large, and traditional multi-agent reinforcement learning (MARL) techniques often become computationally intractable; see [Busoniu et al., 2008, Yang and Wang, 2020, Zhang et al., 2021, Gronauer and Diepold, 2022, Wong et al., 2023] for recent reviews.

Mean Field Games (MFGs) [Lasry and Lions, 2007, Huang et al., 2006] offer a principled framework for approximating such large-scale systems by modeling the interaction between a single representative agent and an evolving population distribution. Drawing on tools from statistical physics and optimal control, MFGs reduce the dimensionality of the problem by considering the limiting behavior as the number of agents tends to infinity. At equilibrium, each agent solves a Markov decision process (MDP) given the population distribution, and the distribution itself must evolve consistently with

---

[*]Work done during period at NYU Shanghai Center for Data Science and the NYU-ECNU Institute of Mathematical Sciences at NYU Shanghai. Contacts: `lm2183@cam.ac.uk`, `kshao@kth.se`, `zdwu@ucla.edu`, `shen.patrick.jiacheng@nyu.edu`, `mathieu.lauriere@nyu.edu`.

[†]Work done during period at NYU Shanghai

39th Conference on Neural Information Processing Systems (NeurIPS 2025).

the agents' policies. Since individual agents are infinitesimal in the limit, they do not affect the population, allowing each agent to ignore second-order feedback effects.

This approximation is particularly relevant for applications involving large populations and continuous state-action spaces, such as economics [Lachapelle et al., 2016, Achdou et al., 2022], finance [Carmona et al., 2017, Cardaliaguet and Lehalle, 2018, Carmona, 2021], engineering [Djehiche et al., 2017], crowd motion [Lachapelle and Wolfram, 2011, Djehiche et al., 2017, Achdou and Lasry, 2018], flocking and swarming [Fornasier and Solombrino, 2014, Nourian et al., 2010], cloud computing [Hanif et al., 2015, Mao et al., 2022], and telecommunication networks [Yang et al., 2016, Ge et al., 2019]. In many of these domains, both the state dynamics and control policies are naturally continuous, and the population distribution often evolves over time rather than remaining stationary.

While classical numerical solvers for MFGs can handle low-dimensional problems in simple domains [Achdou and Laurière, 2020], they are limited by the curse of dimensionality and do not scale to complex or high-dimensional settings. Deep learning methods have been proposed to solve high-dimensional problems (see e.g. [Hu and Laurière, 2024] for an overview) but these methods generally struggle to solve MFGs in very complex environments. To address these limitations, recent work has turned to model-free reinforcement learning (RL) as promising approaches to solving MFGs [Guo et al., 2019, Subramanian and Mahajan, 2019, Elie et al., 2020, Fu et al., 2019, Cui and Koeppl, 2021, Angiuli et al., 2022, Yardim et al., 2023, Ocello et al., 2024]; see [Laurière et al., 2022b] for a recent survey. However, most existing methods are restricted to **finite state and action spaces** and **stationary population distributions**. Extending RL frameworks to continuous-space, non-stationary MFGs presents significant challenges, especially for learning time-dependent population dynamics and solving the resulting fixed-point problems. In contrast to MDPs, where the goal is to optimize a single agent's trajectory, solving MFGs requires learning both an optimal response and a consistent population evolution. **To the best of our knowledge, no existing RL algorithms are capable of learning the solution of non-stationary MFGs with continuous state and action spaces**.

**Contributions.** This paper introduces Density-Enhanced Deep-Average Fictitious Play (DEDA-FP) (see Figure 1), a novel deep reinforcement learning (DRL) algorithm for **non-stationary** Mean Field Games (MFGs) with **continuous** state and action spaces. Our approach extends **Fictitious Play (FP)**, a classical game-theoretic learning scheme that iteratively updates each agent's policy to optimally respond to the evolving population behavior.

To address the challenge of averaging neural policies, we use **DRL** (Soft Actor-Critic and Proximal Policy Optimization) to compute approximate best responses and **supervised learning** to represent the averaged policy across FP iterations. This hybrid strategy ensures scalability and accurate policy approximation.

We also train a time-dependent Conditional Normalizing Flow (CNF) to model the non-stationary evolution of the **population distribution**, enabling **sampling** from the equilibrium mean field and **density estimation**. This model accurately captures MFGs with local dependence on population density, unlike empirical distributions, and **improves sampling time efficiency tenfold** compared to our benchmarks.

We validate our method with three experiments of increasing complexity and provide an error propagation analysis (Theorem 1). Our contributions address key challenges in applying RL to MFGs, including **time-dependence**, **continuous spaces**, and **local density effects**, representing a significant step toward scalable, model-free solutions for real-world multi-agent systems.

## 1.1 Related Work

The following works are particularly relevant to our context and help clarify our methodological contributions (see Table 1 for a summary). [Perrin et al., 2021] develop a DRL algorithm based on **Fictitious Play (FP)** for continuous state and action spaces. However, their focus is restricted to **stationary MFGs** (i.e., time-invariant mean fields), and their method does *not* learn the Nash equilibrium policy. Instead, it learns a collection of best responses from which a player may sample to imitate the average behavior. [Laurière et al., 2022a] propose two DRL algorithms and in particular a variant of FP which does learn the **Nash equilibrium policy**. While we draw inspiration from their approach to represent the average policy, their method is limited to **discrete state and action spaces**.

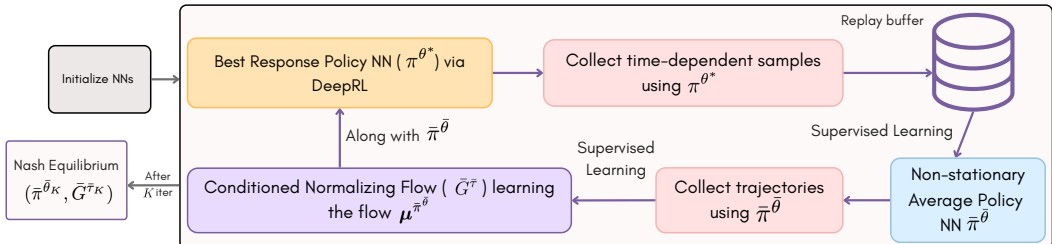

Figure 1: Overview of our **DEDA-FP** model. Our framework uses three main steps, built upon the Fictitious Play algorithm, to fully solve the MFG problem (details in Section 3): (1) computation of the best response using **Deep RL algorithms**; (2) learning a **policy neural network** to approximate the average policy over past policies; and (3) learning a **Time-Conditioned Normalizing Flow** to approximate the average distribution over past mean-field flows.

[Zaman et al., 2020] tackle **non-stationary MFGs** using actor-critic methods in a discrete-time **linear-quadratic (LQ)** setting. Although their model operates in continuous state and action spaces, it is confined to the LQ regime, where optimal policies are deterministic and linear. [Angiuli et al., 2023] study DRL algorithms for MFGs with continuous states and actions, and include a generative model for the population distribution. However, they only address **stationary LQ problems**, and their generative model can produce samples from the mean field but *cannot* estimate the **density at a given location**, making it unsuitable for models with local mean field dependence.

| Method | Cont. space | General $r, P$ | NE policy | Local. dep. | Non-stat. |
|---|---|---|---|---|---|
| **DEDA-FP** | ✓ | ✓ | ✓ | ✓ | ✓ |
| Zaman et al. [2020] | ✓ | ✗ | ✓ | ✗ | ✓ |
| Perrin et al. [2021] | ✓ | ✓ | ✗ | ✓ | ✗ |
| Laurière et al. [2022a] | ✗ | ✓ | ✓ | ✓ | ✓ |
| Angiuli et al. [2023] | ✓ | ✗ | ✓ | ✗ | ✗ |

Table 1: Comparison between our approach and related works. Our approach is the first to learn the Nash equilibrium policy and distribution for continuous space non-stationary MFGs with general dynamics and rewards, including possibly local dependence on the mean field.

## 2 Non-stationary continuous MFGs

**Notations.** Let $\mathcal{X}$ and $\mathcal{A}$ be respectively the state and action spaces, which can be continuous. To fix the ideas, we will take $\mathcal{X} = \mathbb{R}^d$ and $\mathcal{A} = \mathbb{R}^k$, where $d$ and $k$ are the respective dimensions. Let $\mathcal{P}(\mathcal{X})$ and $\mathcal{P}(\mathcal{A})$ denote the sets of probability distributions on $\mathcal{X}$ and $\mathcal{A}$, respectively. Let $N_T$ be the number of time steps. We will use bold symbols for sequences, i.e., functions of time. The state distribution of the population at time $t$ is called the **mean field** and will be denoted by $\mu_t \in \mathcal{P}(\mathcal{X})$. We denote by $\mu_0$ the initial distribution, which is assumed to be fixed and known from the players.

**Dynamics.** To define the mean field game, we first need to define the dynamics of the state $x_t$ of a representative player when the mean field sequence $\boldsymbol{\mu} = (\mu_t)_{t=0,\ldots,N_T}$ is given. We consider a general dynamics: if the agent is in state $x_t$, takes action $a_t$ and the mean field is currently $\mu_t$, then the next state is sampled according to:

$$x_{t+1} \sim P_t(\cdot|x_t, a_t, \mu_t), \tag{1}$$

where $P : \{0, \ldots, N_T\} \times \mathcal{X} \times \mathcal{A} \times \mathcal{P}(\mathcal{X}) \to \mathcal{P}(\mathcal{X})$ is the *transition kernel*. A typical setting is when the transitions are given by a transition function, namely, $x_{t+1} = F_t(x_t, a_t, \mu_t, \epsilon_t)$, where $F : \{0, \ldots, N_T\} \times \mathcal{X} \times \mathcal{A} \times \mathcal{P}(\mathcal{X}) \to \mathcal{X}$ is the *transition function* and $(\epsilon_t)_{t \geq 0}$ is a sequence of i.i.d. noises. A typical example is the time-discretization of a continuous time stochastic differential equation (SDE): $\mathcal{X} = \mathbb{R}^d, \mathcal{A} = \mathbb{R}^k$ and $F_t(x_t, a_t, \mu_t, \epsilon_t) = x_t + b_t(x_t, a_t, \mu_t) + \sigma\epsilon_t$, where $b : \{0, \ldots, N_T\} \times \mathcal{X} \times \mathcal{A} \times \mathcal{P}(\mathcal{X}) \to \mathcal{X}$ is the *drift*, and $\sigma$ is the *volatility*. This setting is particularly relevant because most of the MFG literature focuses on SDE-type dynamics, see e.g. [Carmona and Delarue, 2018]. But we stress that here, only time is discretized; space is continuous, in contrast with

most of the literature on RL for MFGs, see e.g. [Laurière et al., 2022b]. From the dynamics of the individual player, we can deduce a dynamics for the whole population, i.e., a transition from $\mu_t$ to $\mu_{t+1}$, as explained below after introducing the notion of policy.

**Policies.** A policy $\boldsymbol{\pi}$ for an individual player is a function from $\{0, \ldots, N_T\} \times \mathcal{X}$ to $\mathcal{P}(\mathcal{A})$ and $\pi_t(\cdot|x) = \pi(\cdot|t, x)$ is the distribution used to pick the next action when time is $t$ and the player's state is $x$. As is common in the MFG literature [Guo et al., 2019, Elie et al., 2020, Cui and Koeppl, 2021, Guo et al., 2023], we consider here decentralized policies, which are functions of the individual state and not of the population distribution. This is because, at equilibrium, the mean field is completely determined by the policy and the initial population distribution, which is assumed to be known. If the whole population uses the same policy $\boldsymbol{\pi}$, this induces a mean field flow $\boldsymbol{\mu^\pi} = (\mu_t^\pi)_{t \geq 0}$, which is determined by the evolution: $\mu_0^\pi = \mu_0$, and for $t = 0, \ldots, N_T - 1$,

$$\mu_{t+1}^{\boldsymbol{\pi}}(x') = \int_{\mathcal{X} \times \mathcal{A}} \mu_t^{\boldsymbol{\pi}}(x)\pi_t(a|x)P(x'|x, a, \mu_t^{\boldsymbol{\pi}})dxda, \quad x' \in \mathcal{X}. \tag{2}$$

**Rewards.** The reward is a function $r : \mathcal{X} \times \mathcal{A} \times \mathcal{P}(\mathcal{X}) \to \mathbb{R}$. When the mean field flow is given by $\boldsymbol{\mu} = (\mu_t)_{t \geq 0}$, the representative agent aims to find a policy $\boldsymbol{\pi} = (\pi_t)_{t \geq 0}$ that maximizes the total expected reward: with $(\mu_t^{\boldsymbol{\pi}})_{t \geq 0}$ is the distribution flow induced by $\boldsymbol{\pi}$, see (2),

$$J_{\boldsymbol{\mu}}(\boldsymbol{\pi}) = \mathbb{E}_{x_t, a_t}\Big[\sum_{t=0}^{N_T} r(x_t, a_t, \mu_t)\Big] = \sum_{t=0}^{N_T} \int_{\mathcal{X} \times \mathcal{A}} \mu_t^{\boldsymbol{\pi}}(x)\pi_t(a|x)r(x, a, \mu_t)dxda.$$

We will use the notations $J_{\boldsymbol{\mu}}(\boldsymbol{\pi})$ and $J(\boldsymbol{\pi}, \boldsymbol{\mu})$ interchangeably.

**Nash equilibrium.** We focus on the notion of Nash equilibrium, in which no player has any incentives to deviate unilaterally. It is defined as follows in the mean field setting.

**Definition 1.** *A **mean-field Nash equilibrium (MFNE)** is a pair $(\boldsymbol{\mu^*}, \boldsymbol{\pi^*}) = (\mu_t, \pi_t)_{t \geq 0}$ of a sequence of population distributions and policies that satisfies:*

1. *$\boldsymbol{\pi^*}$ maximizes $\boldsymbol{\pi} \mapsto J_{\boldsymbol{\mu^*}}(\boldsymbol{\pi})$;*
2. *For every $t \geq 0$, $\mu_t^*$ is the distribution of $x_t$ given by dynamics (1) with $(\boldsymbol{\pi^*}, \boldsymbol{\mu^*})$.*

It can be expressed more concisely using the notion of exploitability, which quantifies how much a player can be better of by deviating from the policy used by the rest of the players.

**Definition 2.** *The **exploitability** of a policy $\boldsymbol{\pi}$ is defined as: $\mathcal{E}(\boldsymbol{\pi}) = \max_{\boldsymbol{\pi'}} J_{\boldsymbol{\mu^\pi}}(\boldsymbol{\pi'}) - J_{\boldsymbol{\mu^\pi}}(\boldsymbol{\pi})$.*

Then, $(\boldsymbol{\pi}, \boldsymbol{\mu^\pi})$ is Nash equilibrium if and only if $\mathcal{E}(\boldsymbol{\pi}) = 0$. The notion of exploitability has been used in several works to assess the convergence of RL algorithms, see e.g. [Perrin et al., 2020].

We discuss our assumption in App. A. Continuous space implies that the mean field, in $\mathcal{P}(\mathcal{X})$, is a priori infinite-dimensional. Furthermore, the non-stationarity of the model ensures that the policy should be time-dependent, departing from most of the RL literature. Another difficulty is that we want to compute the equilibrium policy, which is in general, a mixed policy, not always learned correctly by algorithms such as FP. Although ad hoc methods have been proposed for linear dynamics and quadratic rewards, an RL algorithm for general models is still lacking.

## 3 Solving Continuous Mean Field Games

Solving a mean field game necessitates finding the flow $(\boldsymbol{\mu^*}, \boldsymbol{\pi^*}) = (\mu_t, \pi_t)_{t \geq 0}$ that satisfies Def. 1. To achieve this, we employ a version of **FP** algorithm. FP was originally introduced in [Brown, 1951] and adapted to MFGs in [Cardaliaguet and Hadikhanloo, 2017] and [Perrin et al., 2020] respectively in continuous time and discrete time MFGs. Our approach relies on computing the best response against the weighted average of previous distributions and updated the distribution consequently (see details in Appendix C). **However, in continuous action spaces, calculating the best response, learning the average policy, and determining the average distribution present significant challenges due to the problem's infinite dimensionality.** To illustrate the effectiveness of our approach, we proceed with a *step-by-step construction*: first, we present a basic algorithm (Algo. 1), followed by a demonstration of the benefits of learning the equilibrium policy (Algo. 2). Finally, we detail our approach, named **DEDA-FP** (Algo. 3), which overcomes the limitations highlighted by these preceding methods.

**Algo. 1: Simple Approach.** As a first method, we implement a simple version of FP, in the spirit of Perrin et al. [2021]. At each iteration of FP, the agent learns the best response against the approximated average population distribution using DRL algorithms such as Soft Actor-Critic (SAC) or Proximal Policy Optimization (PPO). Subsequently, this policy is added to a behavioral buffer. The population state at iteration $k$ is approximated by simulating $N-1$ trajectories, each generated by one of $N-1$ policies sampled uniformly from the buffer. Note that this constitutes a two-level approximation: first, we approximate the average policy by sampling from the buffer, and then we mimic the population distribution by sampling trajectories. Details are provided in Appendix C. **However, at the conclusion of this algorithm, we do not obtain either the Nash equilibrium policy or the mean field distribution.**

**Algo. 2: Learning the Nash equilibrium policy.** To address the main limitation of the simple approach, we train a policy network to learn the average policy. To this end, we draw inspiration from [Heinrich and Silver, 2016, Laurière et al., 2022a] and use supervised learning with a suitably defined replay buffer. We present here the main ideas and the details are provided in Appendix C. Concretely, at every iteration we collect samples from the best response and we train a neural network (NN) to minimize the Negative Log-Likelihood:

$$\mathcal{L}_{\text{NLL}}(\theta) = \mathbb{E}_{(t,s,a) \sim \mathcal{M}_{SL}} \left[ -\log \pi^\theta(a|t,s) \right] = -\frac{1}{M} \sum_{i=1}^{M} \log \mathcal{N}(a_i; \mu_\theta(s_i, t_i), \sigma_\theta(s_i, t_i))$$

where $\mathcal{N}(\cdot)$ is the Gaussian probability density function, $M$ is the size of the replay buffer $\mathcal{M}_{SL}$ containing all the triples $(t, s, a)$ sampled from the previous policies and $\mu_\theta$ and $\sigma_\theta$ are the mean and standard deviation predicted by the policy network for the state $s_i$ at time $t_i$. Furthermore, this approach allows us, at iteration $k$, to compute the best response against $N-1$ agents using the learned average policy, thereby avoiding the need to sample uniformly from the behavioral buffer. Further details are provided in Appendix C.

> **Remaining Challenges.** At this point, the algorithm can learn the **equilibrium policy**, which is one part of the solution to the MFGs (see Def. 1). However, the other part, namely the **equilibrium mean field**, is still lacking. Most existing works then approximate the optimal mean field distribution by sampling a large number of trajectories to adequately cover the state space. However, as we will elaborate in Sec. 5, there are several key limitations to this approach:
>
> **1.** Many mean field games derive their complexity and richness from **local interactions**, where the dynamics or rewards depend on the population density (e.g., congestion, entropy maximization). Without a direct model for the mean field distribution, the density must be estimated indirectly (e.g., via Gaussian convolution), which can alter the nature of the problem.
>
> **2.** In the **evaluation** or rollout phase, estimating the mean field and its density requires sampling many trajectories at each step. This becomes computationally expensive, especially in state spaces of dimension $d \geq 2$, and can significantly slow down execution.

For the reasons mentioned above, we propose a novel algorithm, **Density-Enhanced Deep Average Fictitious Play** solver (DEDA-FP), that fully solves MFGs by learning both the Nash equilibrium policy and the mean field distribution, enabling us to both sample from it and the compute its density.

### 3.1 DEDA-FP

Our method incorporates best response computation using deep reinforcement learning and learns the average policy with supervised learning, as depicted in the previous scheme. Furthermore, it completely solves the mean field game problem by learning the Nash equilibrium mean field distribution. Inspired by and extending the approach of Perrin et al. [2021], we utilize a time-conditioned generative model to learn the mean field flow $\boldsymbol{\mu}^{\boldsymbol{\pi}} = (\mu_t^{\boldsymbol{\pi}})_{t \geq 0}$. Specifically, we employ a Conditional Normalizing Flow [Winkler et al., 2019, Rezende and Mohamed, 2015, Kobyzev et al., 2020] which is a particular generative model that learns a complex probability distribution $p(\mathbf{x}|t)$ conditioned on a time variable $t \in [0, T]$. It achieves this by transforming a simple base distribution $p_0(\mathbf{z})$ (e.g., a standard Gaussian) into the target distribution $p(\mathbf{x}|t)$ through a sequence of invertible transformations $f_1, f_2, \ldots, f_K$, where each transformation is conditioned on the time $t$. The probability density of a sample $\mathbf{x}$ from the target distribution $p(\mathbf{x}|t)$ is computed as:

$$p(\mathbf{x}|t) = p_0(f^{-1}(\mathbf{x}, t)) \left| \det \left( \frac{\partial f^{-1}(\mathbf{x}, t)}{\partial \mathbf{x}} \right) \right|,$$

where $f^{-1}(\mathbf{x}, t) = f_K^{-1}(f_{K-1}^{-1}(\ldots f_1^{-1}(\mathbf{x}, t) \ldots))$ is the inverse of the entire flow, and $\det\left(\frac{\partial f^{-1}(\mathbf{x}, t)}{\partial \mathbf{x}}\right)$ is the determinant of the Jacobian of the inverse transformation.

The conditioning on time $t$ is typically incorporated into the parameters of the transformation functions $f_k$. For example, if $f_k$ is an affine transformation, its parameters (e.g., the scaling and translation) would be functions of $t$. Similarly, for more complex flow architectures like neural spline flows, the parameters of the splines would be conditioned on $t$ (see Sec. 5 for more details about the architecture we used).

To learn the parameters of the Conditional Normalizing Flow, we employ Maximum Likelihood Estimation (MLE). This is equivalent to minimizing the Negative Log-Likelihood (NLL) of the observed data. Given a dataset of $N$ samples $\{\mathbf{x}_i, t_i\}_{i=1}^N$ drawn from the time-dependent distribution, the NLL loss function is given by:

$$\mathcal{L}_{\text{NLL}} = -\frac{1}{N} \sum_{i=1}^N \left[ \log p_0(f^{-1}(\mathbf{x}_i, t_i)) + \log \left| \det\left(\frac{\partial f^{-1}(\mathbf{x}_i, t_i)}{\partial \mathbf{x}_i}\right) \right| \right].$$

By learning the parameters of these time-conditioned transformations using maximum likelihood estimation on a dataset of time-dependent distributions, the model learns to generate samples from and estimate the probability density of $p(\mathbf{x}|t)$ for any $t \in [0, T]$. Algo. 3 summarizes our method. For notations and further discussion on the choices of the individual components refer to App. C.

---

**Algo. 3** Density-Enhanced Deep Average Fictitious Play (**DEDA-FP**)

---

1: **Input:** $\mu_0$: initial distribution; $N_{sa}$: number of state-action pairs to collect at every iteration, $N$: population size in population simulation; $K$: number of iterations.
2: **Initialize:** $(\theta_0^* = \bar{\theta}_0, \bar{\tau}_0)$ at random; empty replay buffer $\mathcal{M}_{SL}$ for supervised learning of average policy; using $\boldsymbol{\pi}_0^* := \boldsymbol{\pi}^{\theta_0^*}$, sample $N_{sa}$ triples $(0, s, a)$ and store them in $\mathcal{M}_{SL}$.
3: **for** iteration $k = 1$ to $K$ **do**
4:    Find the best response $\boldsymbol{\pi}_k^* := \boldsymbol{\pi}^{\theta_k^*}$ vs the $N-1$ agents using $\bar{\boldsymbol{\pi}}_{k-1} := \bar{\boldsymbol{\pi}}^{\bar{\theta}_{k-1}}$ using **Deep RL**:
$$\boldsymbol{\pi}_k^* = \arg\max_{\boldsymbol{\pi}} J_{\mu_0}^N(\boldsymbol{\pi}, \bar{\mathbf{G}}_{k-1})$$
5:    Collect $N_{sa}$ time-state-action samples of the form $(t, s, a)$ using $\boldsymbol{\pi}_k^*$ and store in $\mathcal{M}_{SL}$.
6:    Train the **NN policy** $\bar{\boldsymbol{\pi}}_k := \bar{\boldsymbol{\pi}}^{\bar{\theta}_k}$ using supervised learning to minimize the categorical loss:
$$\mathcal{L}_{\text{NLL}}(\bar{\theta}) = \mathbb{E}_{(t,s,a) \sim \mathcal{M}_{SL}} \left[ -\log \bar{\boldsymbol{\pi}}^{\bar{\theta}}(a|t, s) \right]$$
7:    Train a **Conditional Normalizing Flow** $\bar{\mathbf{G}}_k := \bar{\mathbf{G}}^{\bar{\tau}_k}$ for the time-dependent mean field $\boldsymbol{\mu}^{\bar{\boldsymbol{\pi}}_k}$ using trajectories generated by $\bar{\boldsymbol{\pi}}_k$ and initialization $\bar{\mathbf{G}}_{k-1}$.
8: **end for**
9: **return** $\bar{\boldsymbol{\pi}}_K, \bar{\mathbf{G}}_K$

---

## 4 On the convergence of DEDA-FP

We analyze the convergence of the DEDA-FP algorithm by extending the theoretical framework developed by [Elie et al., 2020]. Our goal is to show that the exploitability of the learned policy converges towards a value determined by the sum of three specific accumulated errors, thus establishing convergence to an $\epsilon$-Nash Equilibrium.

We first formalize the **error sources** based on distance $d_{N_T}$ between sequences of mean fields and distance $d_\Pi$ between policies (see App. B for details).

1. **Best Response Error.** The sub-optimality of the DRL policy $\boldsymbol{\pi}_k^*$ wrt the mean-field flow $\bar{\mathbf{G}}_K$ from the previous iteration: $\epsilon_{br}^k := J(\text{BR}(\bar{\mathbf{G}}_{k-1}), \bar{\mathbf{G}}_{k-1}) - J(\boldsymbol{\pi}_k^*, \bar{\mathbf{G}}_{k-1}) \geq 0$.

2. **Average Policy Error.** The error in the supervised learning step: $\epsilon_{sl}^k := d_\Pi(\bar{\boldsymbol{\pi}}_k, \boldsymbol{\Pi}_k^{\text{true}})$ where $\boldsymbol{\Pi}_k^{\text{true}} := \frac{1}{k} \sum_{i=1}^k \boldsymbol{\pi}_i^*$ is the true average policy.

3. **Distribution Error.** The error of the CNF model $\bar{\mathbf{G}}_k$ in approximating the true mean-field flow $\boldsymbol{\mu}^{\bar{\boldsymbol{\pi}}_k}$: $\epsilon_{cnf}^k := d_{N_T}(\bar{\mathbf{G}}_k, \boldsymbol{\mu}^{\bar{\boldsymbol{\pi}}_k})$.

We will use the following two assumptions, which are satisfied under mild conditions on the reward and transition functions:

**Assumption 1** (Lipschitz continuity of $J$). *For any policy $\boldsymbol{\pi}$ and any two flows $\boldsymbol{\mu}_1, \boldsymbol{\mu}_2$: $|J(\boldsymbol{\pi}, \boldsymbol{\mu}_1) - J(\boldsymbol{\pi}, \boldsymbol{\mu}_2)| \leq L \cdot d_{N_T}(\boldsymbol{\mu}_1, \boldsymbol{\mu}_2)$.*

**Assumption 2** (Lipchitz continuity of MF). *For any policies $\boldsymbol{\pi}_1, \boldsymbol{\pi}_2$, the generated mean fields satisfy: $d_{N_T}(\boldsymbol{\mu}^{\boldsymbol{\pi}_1}, \boldsymbol{\mu}^{\boldsymbol{\pi}_2}) \leq L_{MF} d_{\Pi}(\boldsymbol{\pi}_1, \boldsymbol{\pi}_2)$.*

The following result provides a bound on the exploitability of the policy computed by our algorithm.

**Theorem 1** (Convergence to approximate Nash equilibrium). *Let $e_k^{true} := J(BR(\boldsymbol{\mu}^{\bar{\boldsymbol{\pi}}_k}), \boldsymbol{\mu}^{\bar{\boldsymbol{\pi}}_k}) - J(\bar{\boldsymbol{\pi}}_k, \boldsymbol{\mu}^{\bar{\boldsymbol{\pi}}_k}) \geq 0$ be the **true exploitability** at iteration $k$, which measures the incentive to deviate from the policy $\bar{\boldsymbol{\pi}}_k$ in the true distribution it generates, $\boldsymbol{\mu}^{\bar{\boldsymbol{\pi}}_k}$. Under Assumptions 1 and 2, we have:*

$$
e_k^{true} < C_0 e_0^{cnf} + \frac{1}{k} \sum_{i=1}^{k-1} \left[ (i+1)\epsilon_{br}^{i+1} + C_1(\epsilon_{sl}^{i+1} + \epsilon_{cnf}^{i+1}) + \frac{C_2}{i} \right]
$$

*for some constants $C_0, C_1, C_2 > 0$.*

The proof is provided in App. B. It relies on analyzing the propagation of errors.

## 5 Experiments

To validate our method, we present three distinct scenarios, each designed to showcase specific strengths and potential applications. The initial two examples illustrate the benefit of learning the average policy without performance degradation, while also revealing limitations in capturing local dependencies. We then introduce a more complex case study to demonstrate the effectiveness of DEDA-FP against established benchmarks, highlighting its ability to directly learn the environment and achieve a richer problem representation. Further results and an additional example (the *price impact model*) can be found in Appendix E.

**Evaluation:** To evaluate the algorithms' performance, we approximate the exploitability (defined in Def. 2). Since the model-free setting prevents direct computation of the optimal value we approximate the first term . In all cases, we conduct 4 independent runs and report the mean and standard deviation of the exploitability across these runs to assess the consistency of our results.

**Architecture details**: For the DRL algorithms, we used the Stable Baselines library by Hill et al. [2018] for the implementation of the DeepRL algorithm (SAC for the first two examples, PPO for the case study). We used an RTX4090 GPU with 24 GB RAM for each experiment. For the policy network we use a multi-layer perceptron (MLP) consisting of two hidden layers, each with 256 units and two parallel output layers for mean and standard deviation. To model distributions, we adapt a version of Neural Spline Flows (NSF) with autoregressive layers [Durkan et al., 2019] for handling time dependencies. More details about NSF work and implementation see Appendix D.

### 5.1 Beach Bar Problem

**Environment:** *"I would like to go to that nice bar on the beach, unless it is too crowded!"*. We consider a continuous space version of the beach bar problem introduced in [Perrin et al., 2020] in discrete space. We take $\mathcal{X} = [0, 1]$, $\mathcal{A} = [-0.3, 0.3]$ as a continuous state space with dynamics: $x_{t+1} = x_t + a_t + \epsilon_t$. If the agent reaches the boundary of $\mathcal{X}$ and tries to exit, it is pushed back inside. $\epsilon_t$ represents the noise and is uniformly distributed over $[-0.1, 0.1]$. We take the initial distribution $\mu_0 = Unif[0, 1]$. The time horizon is set as $N_T = 10$. For the one-step reward function, we take $r(x, a, \mu) = -C_1|x - x_{\text{bar}}|^2 - C_2\mu(x) - C_3|a|^2$, where $C_1, C_2, C_3 \geq 0$, $x_{bar} = 0.5$ and $\mu(x)$ denotes the value of the density at $x$ and represent the *congestion avoidance*.

**Numerical results:** Fig. 2 presents a comparison between the algorithms, illustrating the final flow $\boldsymbol{\mu}^{\bar{\boldsymbol{\pi}}_K}$ after the last Fictitious Play iteration ($K$). As can be seen in the figure, **DEDA-FP** demonstrates a superior interpretation of the problem formulation, achieving a smoother distribution concentrated around the center. Furthermore, the exploitability decay analysis indicates that this improved distributional representation in Algo. 3 does not come at the cost of performance. An important challenge highlighted by this problem, revealing a limitation of both benchmarks Algo. 1

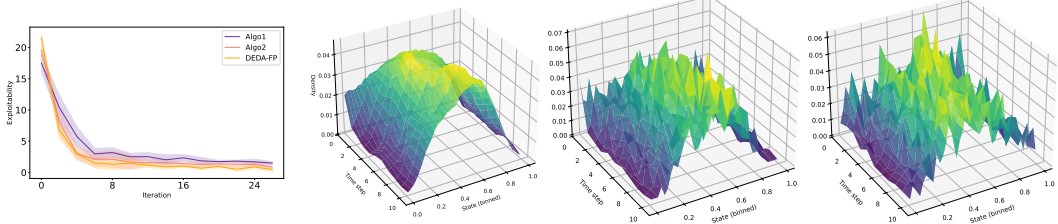

Figure 2: **Beach Bar Problem.** a): exploitability of Algo. 1, Algo. 2 and DEDA-FP; b) NE distribution DEDA-FP; c) NE distribution Algo. 2; d) NE distribution Algo. 1

and Algo. 2, lies in the approach to compute the local dependence $\mu(x)$ within the reward function. Due to the absence of an accessible approximation model for querying, we redefined the mean field cost as follows: $\mu(x) = (\mu^N * \rho)(x)$, and $(\mu * \rho)(x) := \int_{\mathcal{X}} \mu(y)\rho(x - y)dy$, $\rho$ is the Gaussian density $\rho(x) := \frac{1}{\sqrt{2\pi\sigma^2}}e^{\frac{-x^2}{2\sigma^2}}$ and $\mu_t^N := \frac{1}{N}\sum_{i=1}^{N}\delta_{X_t^i}$ where $X_t^i$ is the position at time $t$ of the agent $i$ and $\delta$ is the Dirac delta measure.

## 5.2 Linear-Quadratic (LQ) model

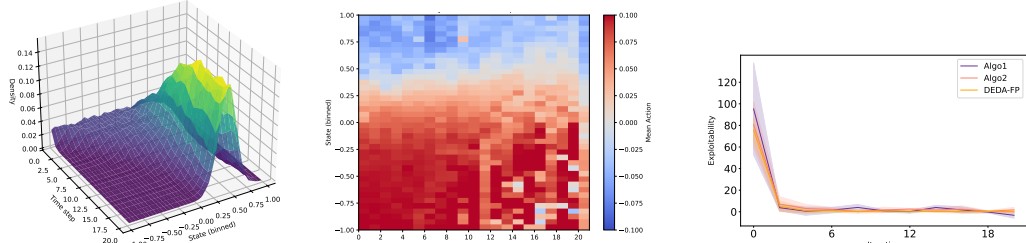

Figure 3: **LQ Model DEDA-FP** Left: mean field flow $\boldsymbol{\mu}^{\bar{\boldsymbol{\pi}}_K}$ at the last Fictitious Play (FP) iteration. Middle: policy $\bar{\boldsymbol{\pi}}_K$ at the last FP iteration. Right: comparison of exploitability between **DEDA-FP** and Algo. 1 and Algo. 2.

**Environment:** We consider a linear-quadratic (LQ) model with continuous state and action spaces and a finite time horizon $N_T$. Similar LQ models have been considered in [Laurière et al., 2022a] with discrete spaces and [Angiuli et al., 2023] with stationary mean field. We take the state space $\mathcal{X} = [-1, 1]$ and the action space $\mathcal{A} = [-0.1, 0.1]$. The dynamics is: $x_{t+1} = Ax_t + Ba_t + \bar{A}\bar{\mu}_t + \epsilon_t$, where $A$, $B$ and $\bar{A}$ are real constants, $\bar{\mu}_t = \int_{\mathcal{X}} x\mu_t(x)\mathrm{d}x$ is the first moment (mean) of the distribution at time $t$, and $\epsilon_t$ represents the noise that is uniformly distributed over $[-0.1, 0.1]$. The reward is: $r(x, a, \mu) = -c_X|x - x_{\text{target}}|^2 - c_A|a|^2 - c_M|x - \bar{\mu}|^2$, where $c_X, c_A$ and $c_M$ are positive constants. The dynamics and the reward are linear and quadratic in the state $x$, the action $a$, and the mean $\bar{\mu}$ of the distribution. The agent learns to maximize the reward by finding an optimal policy that balances staying close to the target state, minimizing the action cost, and remaining near the population mean. In the experiment, we set $N_T = 20$, $A = 1$, $B = 1$, $\bar{A} = 0.06$. The reward coefficients are $c_{\mathcal{X}} = 5$, $c_A = 0.1$, and $c_M = 1$.

**Numerical results:** Fig. 3 shows the mean field flow and the learned policy after the last FP iteration by Algo. 2. The distribution concentrates near the target position $x_{\text{target}} = 0.6$ that aligns with the reward's high weight target discrepancy term. The learned policy also shows its linearity with respect to the state. The agent takes action that converges to the target position with increasing $|a|$ as the distance from the target increases. It is important to note that at later time steps, the policy's predictions far from the target may exhibit inconsistencies. However, this is not a limitation, as it is caused by the low agent density in those regions, rendering the action choices effectively arbitrary at those times. **This example demonstrates the policy network's effectiveness in approximating the average policy, and, importantly, shows that performance is not degraded.** The averaged exploitability curves of Algo. 1 and Algo. 2 both show the exploitability converges to zero quickly after several FP iterations and stay near zero, confirm the latter statement.

## 5.3 Case Study: *4-rooms exploration*

Building upon the limitations observed with Algo. 1 and Algo. 2, we now introduce a more complex setting to further highlight the strengths of the **DEDA-FP** (Algo. 3) against the benchmarks. In this problem, the approximation of the mean-field distribution becomes a critical aspect, primarily due to its inherent nature of entropy maximization.

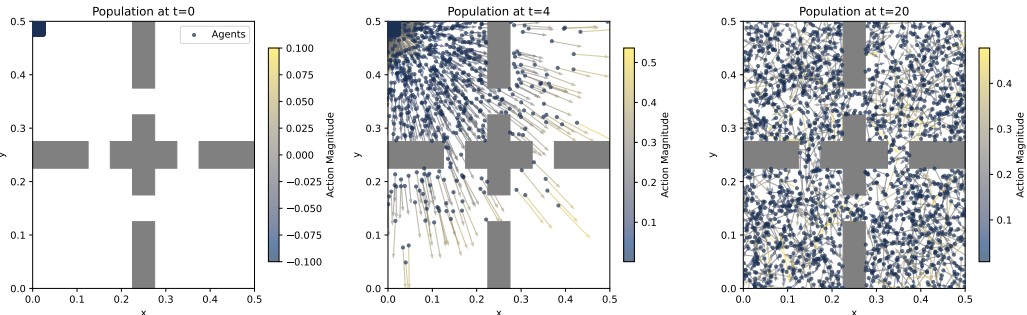

Figure 4: **4-rooms Exploration.** Visualization of a large, finite population of 2000 agents and their velocity vectors during exploration in the 4-rooms environment. The agents' behavior is governed by the mean-field Nash equilibrium policy learned by the DEDA-FP solver, Algo. 3. This shows how well the mean-field approximation captures the behavior of a large-population system.

**Environment:** The 4-rooms exploration MFG has been introduced using discrete spaces in [Geist et al., 2022] and has served as a benchmark e.g. in [Laurière et al., 2022a, Algumaei et al., 2023]. The state space and the action space are 2 dimensional ($d = 2$) and the states have constraints represented by walls forming four connected rooms. While the original model was discrete, we consider here a continuous space generalization, which is more natural because pedestrians move in continuous space. The action is the vector of movement (velocity) and the reward decreases with the mean field density: a larger density at the player's location means a smaller reward. Hence the players are encouraged to move in order to go to less crowded regions. In the end, if the time horizon is long enough, the mean field density becomes uniform. In this example, the stationary distribution is trivial and the key point is to learn the entire flow from the initial distribution to the stationary one. Following, the mathematical formulation. We take $\mathcal{X} = [0, 1]^2$, interpreted as a 2D domain. Time horizon $N_T$ is set to 20. There are obstacles (walls) such that the domain has the shape of four connected rooms (see Fig. 5). The dynamics are: $x_{t+1} = x_t + v_t + \epsilon_t$, except that the agent cannot cross a wall. The reward is: $r(x, v, \mu) = -c_A \|v\|_2^2 - c_M \log(\mu(x) + \epsilon)$, where $c_A, c_M$ and $\epsilon$ are positive constants, with $\epsilon$ very close to 0. **This reward (entropy maximization) discourages the agent from taking large actions (i.e., from moving a lot) and from being at a crowded location**. The initial distribution is uniform over a small square at the top left corner. We expect the agents to spread throughout the domain and, if the time horizon is long enough, the population distribution should converge to the uniform distribution over the domain.

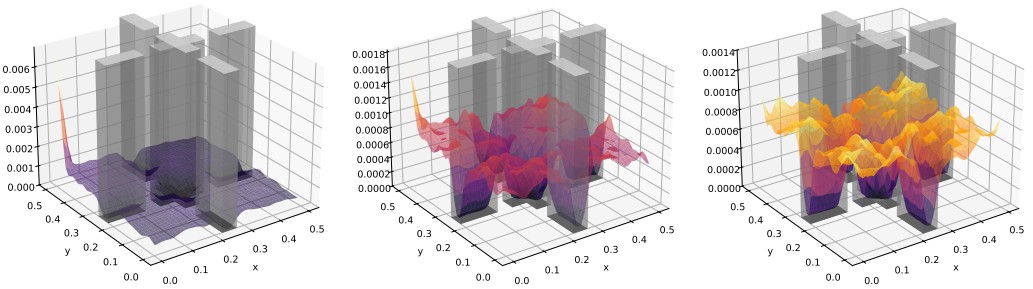

Figure 5: **4-rooms Exploration - NE flow.** The three plots represent the dynamics of the Nash Equilibrium mean field flow $\bar{\mathbf{G}}_K$ at time $t = 6, 15, 20$, obtained by **DEDA-FP**. It can be seen how the population is spreading across the 4 different rooms.

**Discussion:** The aim of this case study was to demonstrate the effectiveness of DEDA-FP (Algo. 3) compared to Algo. 1 and Algo. 2. Figs. 5, 6 and 7 summarize our results. First, DEDA-FP enables direct access to the local dependence $\mu(x)$ in the reward function, without the need to compute the convolution or any other approximation of the local density. Secondly, by leveraging the generative model's sample efficiency, our method yields a superior mean-field distribution representation during training (given a fixed-time budget for all algorithms) while maintaining comparable final performance.

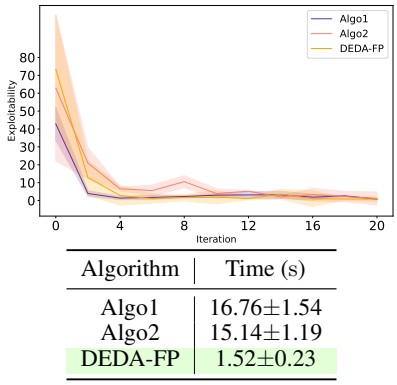

| Algorithm | Time (s) |
|---|---|
| Algo1 | 16.76±1.54 |
| Algo2 | 15.14±1.19 |
| DEDA-FP | 1.52±0.23 |

Figure 6: (top) Exploitability decay. (bottom) Time to sample 5000 trajectories.

*Scalable Sampling:* In Fig. 6 (bottom) we display a table highlighting DEDA-FP's $> 10\times$ efficiency advantage over Algo. 1 and Algo. 2 in generating 5000 trajectories (time horizon: $T = 20$). This represents a substantial reduction in the computational cost of rollouts (critical, for instance, when applying the mean field policy in finite agent settings; see Fig. 4), enabling a considerably faster approximation of the environment (distribution and reward).

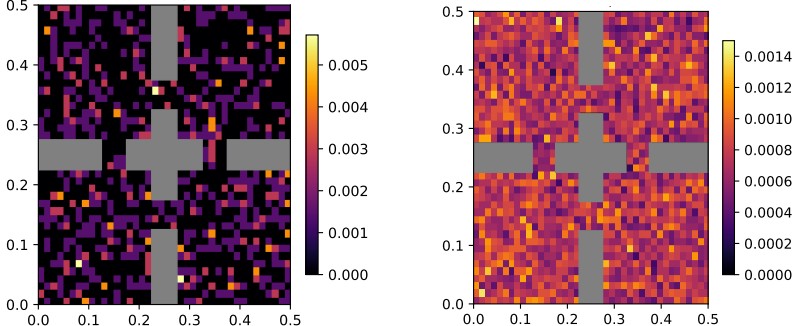

Figure 7: **4-rooms Exploration - Algo 2 vs DEDA-FP** The figure shows the last step Nash equilibrium policy generate by (left) Algo. 2 and (right) **DEDA-FP** with a fixed-time budget of $10s$. Remarkably, DEDA-FP can approximate the mean field distribution by sampling 10 times more agents than Algo. 2.

## 6 Conclusion and Limitations

**Conclusions.** In this paper, we introduce a deep reinforcement learning algorithm for non-stationary continuous MFGs. Our primary contribution lies in the development of DEDA-FP, which combines the strengths of Fictitious Play and supervised learning to accurately compute both the Nash equilibrium policy and the mean field distribution. Our approach enables efficient sampling and density approximation by leveraging time-conditioned normalizing flows, addressing the critical limitations in scalability and local mean field dependencies. In Theorem 1, we provide an error propagation analysis of DEDA-FP. Through three increasingly complex numerical experiments, we demonstrate the effectiveness of DEDA-FP in solving continuous space non-stationary MFGs with general dynamics and rewards. The results show that our approach yields a significant contribution to the application of RL techniques to continuous MFGs.

**Limitations and future work.** As of now, we are still lacking a complete theoretical understanding of the proposed algorithm, particularly due to the complexity of analyzing deep neural networks training. We also left for future work extensions beyond standard MFGs, such as multiple populations and graphon games, or MFGs with common noise and real-world applications. Furthermore, our present evaluation relies on approximate exploitability, which, while a state-of-the-art technique for assessing Nash equilibria, provides an evaluation that is inherently dependent on the environment approximation. Future research will investigate this aspect further.

## Acknowledgments and Disclosure of Funding

M.L. is affiliated with the NYU Shanghai Center for Data Science and the NYU-ECNU Institute of Mathematical Sciences at NYU Shanghai. J.S. is partially supported by NSF Award 1922658. Computing resources were provided by NYU Shanghai HPC.

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

## A  Model Assumptions

We discuss our assumption here.

- **Existence:** Our work assumes the existence of a Mean Field Nash Equilibrium (MFNE). This assumption is grounded in established results for related problems, as existence proofs typically rely on fixed-point arguments under specific regularity conditions on the coefficients not yet proven for our exact setting. Specifically, our justification relies on the work of [Saldi et al., 2020], who established the existence of an equilibrium for a broad class of discrete-time MFGs in an infinite-horizon, risk-sensitive setting. Crucially, their work also provides an argument that this infinite-horizon, risk-sensitive cost can be approximated by a finite-horizon game. Since our finite-horizon setting is a well-posed approximation of a problem where existence is proven, we can assume that an equilibrium also exists in our case. Furthermore, we expect that existence holds under mild assumptions (typically, continuity) using the Schauder fixed point theorem in a suitable functional space. Under more restrictive assumptions (typically, Lipschitz continuity with a small Lipschitz constant), the Banach fixed point theorem yields both existence and uniqueness. While a full proof is beyond the scope of the present paper, which focuses on a DRL algorithm, we included this discussion in Section 2 of the paper, after the definition of MFNE (Definition 1).

- **Uniqueness of Equilibrium:** For our theoretical convergence discussion, we also assume uniqueness. This is not required for the DEDA-FP algorithm itself to run. Following the approach of e.g. [Elie et al., 2020], uniqueness can be ensured by adopting the standard Lasry-Lions monotonicity assumption. Intuitively, this condition means that the reward function discourages agents from being too concentrated.

## B  Proof of the convergence of DEDA-FP

We are going to prove Theorem 1.

As mentioned in Section 4, we follow the methodology developed by [Elie et al., 2020]. Their analysis rigorously bounds the propagation of errors from an approximate best-response computation. We extend this framework to account for two additional sources of error inherent to our algorithm: the average policy error and the distribution error.

We first bound the exploitability with respect to the tractable distribution $\bar{\mathbf{G}}_k$ learned by the algorithm (Lemma 1). Then we connect this result with the true value of the exploitability to show that the pair is an approximate Nash equilibrium $(\bar{\boldsymbol{\pi}}_k, \boldsymbol{\mu}^{\bar{\boldsymbol{\pi}}_k})$ (Lemma 2). Combining these two bounds yield Theorem 1 stated above.

We define $d_{N_T}(\boldsymbol{\mu}_1, \boldsymbol{\mu}_2) = \max_{t=0,\dots,N_T} W_1(\mu_{1,t}, \mu_{2,t})$ and $d_\Pi(\boldsymbol{\pi}_1, \boldsymbol{\pi}_2) := \max_{t=0,\dots,N_T} \int_{\mathcal{X}} W_1(\pi_{1,t}(\cdot|x), \pi_{2,t}(\cdot|x))dx$, where $W_1$ is the Wasserstein-1 distance on $\mathcal{X} = \mathbb{R}^d$ and we consider only a class of regular policies.

**Lemma 1.** *Let Assumptions 1 and 2 above hold. We have:*

$$e_k \le \frac{e_0}{k} + \frac{1}{k}\sum_{i=1}^{k}\left((i+1)\epsilon_{br}^{i+1} + C_1(\epsilon_{sl}^{i+1} + \epsilon_{cnf}^{i+1}) + \frac{C_2}{i}\right)$$

*for some constants $C_1, C_2 > 0$, where $e_k := J(BR(\bar{\mathbf{G}}_k), \bar{\mathbf{G}}_k) - J(\bar{\boldsymbol{\pi}}_k, \bar{\mathbf{G}}_k)$ is the **tractable exploitability** defined with respect to the learned distribution $\bar{\mathbf{G}}_k$.*

*Proof.* The proof adapts the derivation of estimate (8) in Theorem 6 of [Elie et al., 2020]. Let $\hat{e}_k := J(\boldsymbol{\pi}_{k+1}^*, \bar{\mathbf{G}}_k) - J(\bar{\boldsymbol{\pi}}_k, \bar{\mathbf{G}}_k)$. The total tractable exploitability is $e_k = J(BR(\bar{\mathbf{G}}_k), \bar{\mathbf{G}}_k) - J(\boldsymbol{\pi}_{k+1}^*, \bar{\mathbf{G}}_k) + \hat{e}_k = \epsilon_{br}^{k+1} + \hat{e}_k$. We analyze the evolution of the term $(k+1)\hat{e}_{k+1} - k\hat{e}_k$. Following the original proof structure, we can prove that:

$$(k+1)\hat{e}_{k+1} - k\hat{e}_k \le (k+1)\left(J(BR(\bar{\mathbf{G}}_{k+1}), \bar{\mathbf{G}}_{k+1}) - J(\boldsymbol{\pi}_{k+1}^*, \bar{\mathbf{G}}_{k+1})\right).$$

The term in the parentheses is exactly $\epsilon_{br}^{k+1}$. This appears to simplify things, but the derivation used in the original proof is made using exact flows $\boldsymbol{\mu}^{\bar{\boldsymbol{\pi}}_k}$ whereas our value functions are defined with respect

to the tractable flows $\bar{\mathbf{G}}_k$. It can be seen that the extra term's magnitude depends on $d_{N_T}(\bar{\mathbf{G}}_{k+1}, \bar{\mathbf{G}}_k)$. We bound this distance using the triangle inequality and our error definitions:

$$d_{N_T}(\bar{\mathbf{G}}_{k+1}, \bar{\mathbf{G}}_k) \leq d_{N_T}(\bar{\mathbf{G}}_{k+1}, \boldsymbol{\mu}^{\bar{\boldsymbol{\pi}}_{k+1}}) + d_{N_T}(\boldsymbol{\mu}^{\bar{\boldsymbol{\pi}}_{k+1}}, \boldsymbol{\mu}^{\bar{\boldsymbol{\pi}}_k}) + d_{N_T}(\boldsymbol{\mu}^{\bar{\boldsymbol{\pi}}_k}, \bar{\mathbf{G}}_k)$$
$$\leq \epsilon_{cnf}^{k+1} + d_{N_T}(\boldsymbol{\mu}^{\bar{\boldsymbol{\pi}}_{k+1}}, \boldsymbol{\mu}^{\bar{\boldsymbol{\pi}}_k}) + \epsilon_{cnf}^{k}.$$

The term $d_{N_T}(\boldsymbol{\mu}^{\bar{\boldsymbol{\pi}}_{k+1}}, \boldsymbol{\mu}^{\bar{\boldsymbol{\pi}}_k})$ is further bounded by:

$$d_{N_T}(\boldsymbol{\mu}^{\bar{\boldsymbol{\pi}}_{k+1}}, \boldsymbol{\mu}^{\bar{\boldsymbol{\pi}}_k}) \leq d_{N_T}(\boldsymbol{\mu}^{\bar{\boldsymbol{\pi}}_{k+1}}, \boldsymbol{\mu}^{\boldsymbol{\Pi}_{k+1}^{true}}) + d_{N_T}(\boldsymbol{\mu}^{\boldsymbol{\Pi}_{k+1}^{true}}, \boldsymbol{\mu}^{\boldsymbol{\Pi}_k^{true}}) + d_{N_T}(\boldsymbol{\mu}^{\boldsymbol{\Pi}_k^{true}}, \boldsymbol{\mu}^{\bar{\boldsymbol{\pi}}_k})$$
$$\leq \epsilon_{sl}^{k+1} + d_{N_T}(\boldsymbol{\mu}^{\boldsymbol{\Pi}_{k+1}^{true}}, \boldsymbol{\mu}^{\boldsymbol{\Pi}_k^{true}}) + \epsilon_{sl}^{k}.$$

The term $d_{N_T}(\boldsymbol{\mu}^{\boldsymbol{\Pi}_{k+1}^{true}}, \boldsymbol{\mu}^{\boldsymbol{\Pi}_k^{true}})$ corresponds to the change in the exact Fictitious Play update, which is known to be of order $O(1/k)$ (Lemma 5 in [Elie et al., 2020]). Let's denote the stability constant as $C_{FP}$. So, $d_{N_T}(\boldsymbol{\mu}^{\boldsymbol{\Pi}_{k+1}^{true}}, \boldsymbol{\mu}^{\boldsymbol{\Pi}_k^{true}}) \leq C_{FP}/k$. Combining these, we get a bound on the change in our tractable distributions:

$$d_{N_T}(\bar{\mathbf{G}}_{k+1}, \bar{\mathbf{G}}_k) \leq \epsilon_{cnf}^{k+1} + \epsilon_{cnf}^{k} + \epsilon_{sl}^{k+1} + \epsilon_{sl}^{k} + \frac{C_{FP}}{k}.$$

Substituting these bounds back into the recursive inequality for $(k+1)e_{k+1} - ke_k$ introduces additive terms related to $\epsilon_{br}$, $\epsilon_{sl}$, and $\epsilon_{cnf}$. The final form of the recursive inequality becomes:

$$(k+1)e_{k+1} - ke_k \leq (k+1)\epsilon_{br}^{k+1} + C_1(\epsilon_{sl}^{k+1} + \epsilon_{cnf}^{k+1}) + \frac{C_2}{k}.$$

Applying Lemma 9 from [Elie et al., 2020] to solve this recursion yields the result. $\square$

Now, it is necessary to establish a connection between the tractable exploitability (bounded in Lemma 1) and the actual exploitability as defined in (1). The subsequent lemma serves to create this connection.

**Lemma 2.** *Under the same assumptions, the true exploitability $e_k^{true}$ is close to the tractable exploitability $e_k$. Specifically,*

$$|e_k^{true} - e_k| \leq 4L \cdot \epsilon_{cnf}^{k},$$

*where $L$ is the Lipschitz constant of Assumption 1.*

*Proof.* We seek to bound $|e_k^{true} - e_k|$. Using the triangle inequality:

$$|e_k^{true} - e_k| \leq \underbrace{|J(\mathrm{BR}(\boldsymbol{\mu}^{\bar{\boldsymbol{\pi}}_k}), \boldsymbol{\mu}^{\bar{\boldsymbol{\pi}}_k}) - J(\mathrm{BR}(\bar{\mathbf{G}}_k), \bar{\mathbf{G}}_k)|}_{\text{Term 1}} + \underbrace{|J(\bar{\boldsymbol{\pi}}_k, \boldsymbol{\mu}^{\bar{\boldsymbol{\pi}}_k}) - J(\bar{\boldsymbol{\pi}}_k, \bar{\mathbf{G}}_k)|}_{\text{Term 2}}.$$

Let us bound each term separately.

*Term 2 (Bound on Policy Value):* This term is straightforward using Assumption 1.

$$|J(\bar{\boldsymbol{\pi}}_k, \boldsymbol{\mu}^{\bar{\boldsymbol{\pi}}_k}) - J(\bar{\boldsymbol{\pi}}_k, \bar{\mathbf{G}}_k)| \leq L \cdot d_{N_T}(\boldsymbol{\mu}^{\bar{\boldsymbol{\pi}}_k}, \bar{\mathbf{G}}_k)$$
$$\leq L \cdot \epsilon_{cnf}^{k}.$$

*Term 1 (Bound on Best Response Value):* This term requires more care. Let $\boldsymbol{\pi}_1 = \mathrm{BR}(\boldsymbol{\mu}^{\bar{\boldsymbol{\pi}}_k})$ and $\boldsymbol{\pi}_2 = \mathrm{BR}(\bar{\mathbf{G}}_k)$. We want to bound $|J(\boldsymbol{\pi}_1, \boldsymbol{\mu}^{\bar{\boldsymbol{\pi}}_k}) - J(\boldsymbol{\pi}_2, \bar{\mathbf{G}}_k)|$. We add and subtract $J(\boldsymbol{\pi}_1, \bar{\mathbf{G}}_k)$:

$$|J(\boldsymbol{\pi}_1, \boldsymbol{\mu}^{\bar{\boldsymbol{\pi}}_k}) - J(\boldsymbol{\pi}_2, \bar{\mathbf{G}}_k)| \leq |J(\boldsymbol{\pi}_1, \boldsymbol{\mu}^{\bar{\boldsymbol{\pi}}_k}) - J(\boldsymbol{\pi}_1, \bar{\mathbf{G}}_k)| + |J(\boldsymbol{\pi}_1, \bar{\mathbf{G}}_k) - J(\boldsymbol{\pi}_2, \bar{\mathbf{G}}_k)|.$$

The first part is again bounded by Lipschitz continuity: $|J(\boldsymbol{\pi}_1, \boldsymbol{\mu}^{\bar{\boldsymbol{\pi}}_k}) - J(\boldsymbol{\pi}_1, \bar{\mathbf{G}}_k)| \leq L \cdot d_{N_T}(\boldsymbol{\mu}^{\bar{\boldsymbol{\pi}}_k}, \bar{\mathbf{G}}_k) \leq L \cdot \epsilon_{cnf}^{k}$. For the second part, $|J(\boldsymbol{\pi}_1, \bar{\mathbf{G}}_k) - J(\boldsymbol{\pi}_2, \bar{\mathbf{G}}_k)|$, we know by definition of best response that $J(\boldsymbol{\pi}_2, \bar{\mathbf{G}}_k) \geq J(\boldsymbol{\pi}_1, \bar{\mathbf{G}}_k)$. So we need to bound the non-negative quantity $J(\boldsymbol{\pi}_2, \bar{\mathbf{G}}_k) - J(\boldsymbol{\pi}_1, \bar{\mathbf{G}}_k)$. Using the fact that $\boldsymbol{\pi}_1$ is a best response against $\boldsymbol{\mu}^{\bar{\boldsymbol{\pi}}_k}$,

$$0 \leq J(\boldsymbol{\pi}_2, \bar{\mathbf{G}}_k) - J(\boldsymbol{\pi}_1, \bar{\mathbf{G}}_k) = J(\boldsymbol{\pi}_2, \bar{\mathbf{G}}_k) - J(\boldsymbol{\pi}_2, \boldsymbol{\mu}^{\bar{\boldsymbol{\pi}}_k})$$
$$+ J(\boldsymbol{\pi}_2, \boldsymbol{\mu}^{\bar{\boldsymbol{\pi}}_k}) - J(\boldsymbol{\pi}_1, \boldsymbol{\mu}^{\bar{\boldsymbol{\pi}}_k}) + J(\boldsymbol{\pi}_1, \boldsymbol{\mu}^{\bar{\boldsymbol{\pi}}_k}) - J(\boldsymbol{\pi}_1, \bar{\mathbf{G}}_k)$$
$$\leq \left( J(\boldsymbol{\pi}_2, \bar{\mathbf{G}}_k) - J(\boldsymbol{\pi}_2, \boldsymbol{\mu}^{\bar{\boldsymbol{\pi}}_k}) \right) + \left( J(\boldsymbol{\pi}_1, \boldsymbol{\mu}^{\bar{\boldsymbol{\pi}}_k}) - J(\boldsymbol{\pi}_1, \bar{\mathbf{G}}_k) \right).$$

Both remaining terms can be bounded by Assumption 1:

$$J(\boldsymbol{\pi}_2, \bar{\mathbf{G}}_k) - J(\boldsymbol{\pi}_1, \bar{\mathbf{G}}_k) \leq |J(\boldsymbol{\pi}_2, \bar{\mathbf{G}}_k) - J(\boldsymbol{\pi}_2, \boldsymbol{\mu}^{\bar{\boldsymbol{\pi}}_k})| + |J(\boldsymbol{\pi}_1, \boldsymbol{\mu}^{\bar{\boldsymbol{\pi}}_k}) - J(\boldsymbol{\pi}_1, \bar{\mathbf{G}}_k)|$$
$$\leq L \cdot d_{N_T}(\bar{\mathbf{G}}_k, \boldsymbol{\mu}^{\bar{\boldsymbol{\pi}}_k}) + L \cdot d_{N_T}(\boldsymbol{\mu}^{\bar{\boldsymbol{\pi}}_k}, \bar{\mathbf{G}}_k)$$
$$\leq 2L \cdot \epsilon_{cnf}^k.$$

Combining the parts for Term 1, we get $|J(\mathrm{BR}(\boldsymbol{\mu}^{\bar{\boldsymbol{\pi}}_k}), \boldsymbol{\mu}^{\bar{\boldsymbol{\pi}}_k}) - J(\mathrm{BR}(\bar{\mathbf{G}}_k), \bar{\mathbf{G}}_k)| \leq L\epsilon_{cnf}^k + 2L\epsilon_{cnf}^k = 3L \cdot \epsilon_{cnf}^k$.

*Final Bound:* Combining the bounds for Term 1 and Term 2:

$$|e_k^{true} - e_k| \leq (3L \cdot \epsilon_{cnf}^k) + (L \cdot \epsilon_{cnf}^k) = 4L \cdot \epsilon_{cnf}^k.$$

$\square$

This completes the proof of Theorem 1.

## C   Algorithms Details

In the context of MFGs, Fictitious Play (FP) operates by iteratively computing the best response of a representative agent against the distribution induced by the average of past best responses of the entire population. The discrete-time FP process involves several key steps at each iteration $k = 0 \ldots, K$:

1. **Best Response Computation**: An agent finds its best response policy $\boldsymbol{\pi}_k^{BR}$ against the approximated average population distribution $\bar{\mu}_{k-1}$.

2. **Average Policy Update**: The average policy $\bar{\boldsymbol{\pi}}_k$ is computed by averaging the current best response policy with all previous policies. In particular we have $\forall t = 0, \ldots, N_T$

$$\bar{\boldsymbol{\pi}}_{k-1}(\cdot|x, t) = \frac{\sum_{i=0}^{i=k} \pi_i^{BR}(\cdot|x, t) \mu_t^{\boldsymbol{\pi}_k^{BR}}(x)}{\sum_{i=0}^{i=k} \mu_t^{\boldsymbol{\pi}_k^{BR}}(x)}$$

3. **Average Distribution Update**: The average population distribution $\bar{\mu}_k$ is updated by averaging the current population distribution with past distributions. In particular we have,

$$\bar{\mu}_k = \frac{k-1}{k} \bar{\mu}_{k-1} + \frac{1}{k} \boldsymbol{\mu}^{\boldsymbol{\pi}_k^{BR}}$$

It can be seen that $\bar{\mu}_k = \boldsymbol{\mu}^{\bar{\boldsymbol{\pi}}_k}$

At the end of the algorithm, the pair $(\bar{\boldsymbol{\pi}}_K, \bar{\mu}_K)$ represents the Nash equilibrium of the mean-field game problem. In a model-free framework, however, direct analytical computation is not feasible. For this reason, all three steps must be approximated to fully solve the game. While Algo. 1 and Algo. 2 only address parts of this problem, **DEDA-FP** (described in Algo. 3) provides the complete solution.

**Notation.**   Here, we provide the notation used in the pseudocodes.

- $\mathcal{M}$ represent the policy buffer use to store all the best responses during FP.

- $\mu_t^{N,\mathcal{M}}$ is the empirical distribution at time $t$, generated by $N$ agents using a policy assigned uniformly at random from the buffer $\mathcal{M}$.

- $\mu_t^{N,\pi}$ is the empirical distribution, at time $t$, generated by $N$ agents using the policy $\bar{\pi}$.

- $J_{\mu_0,\mathcal{M}}^N(\boldsymbol{\pi})$ is the approximated cost function, computed against $N-1$ trajectories. These trajectories originate from points sampled from the initial distribution $\mu_0$ and subsequently evolve according to $N-1$ policies sampled uniformly from the buffer $\mathcal{M}$.

- $J_{\mu_0,\bar{\pi}}^N(\boldsymbol{\pi})$ is the approximated cost function, computed against $N-1$ trajectories. These trajectories originate from points sampled from the initial distribution $\mu_0$ and subsequently all evolve according to $\bar{\pi}$.

- $\mathcal{L}_{\text{NLL}}(\theta) = \mathbb{E}_{(t,s,a) \sim \mathcal{M}_{SL}} \left[ -\log \boldsymbol{\pi}^{\theta}(a|t,s) \right] = -\frac{1}{M} \sum_{i=1}^{M} \log \mathcal{N}(a_i; \mu_\theta(s_i, t_i), \sigma_\theta(s_i, t_i))$, where $M$ is the size of the replay buffer $\mathcal{M}_{SL}$ containing all the triples $(t, s, a)$ sampled from the previous policies and $\mu_\theta$ and $\sigma_\theta$ are the mean and standard deviation predicted by the policy network for the state $s_i$ at time $t_i$.

---

**Algo. 1** Simple Approach

---

1: **Input:** Initial distribution $\mu_0$; population size $N$; Number of iterations $K$.
2: **Initialize:** $\theta_0^*$, $\mathcal{M} := \{\boldsymbol{\pi}_0^*\}$ where $\boldsymbol{\pi}_0^* := \boldsymbol{\pi}^{\theta_0^*}$.
3: **for** iteration $k = 1$ to $K$ **do**
4:     Using **DRL**, find the best response $\boldsymbol{\pi}_k^* := \boldsymbol{\pi}^{\theta_k^*}$ such that:

$$\boldsymbol{\pi}_k^* = \arg\max_{\boldsymbol{\pi}} J_{\mu_0, \mathcal{M}}^N(\boldsymbol{\pi})$$

5:     Add $\boldsymbol{\pi}_k^*$ in $\mathcal{M}$
6: **end for**
7: **return** $\mathcal{M}$

---

**Algo. 2** Learning the NE Policy

---

1: **Input:** Initial distribution $\mu_0$; $N_{sa}$: number of state-action pairs to collect at every iteration, $N$: population size in population simulation; number of iterations $K$.
2: **Initialize:** $\theta_0^*$ and set $\bar{\theta}_0 = \theta_0^*$ since $(\bar{\boldsymbol{\pi}}^{\bar{\theta}_0} = \boldsymbol{\pi}^{\theta_0^*})$; sample, according to $\boldsymbol{\pi}_0^* := \boldsymbol{\pi}^{\theta_0^*}$, $N_{sa}$ (time)-state-action triples $(0, s, a)$ and define $\mathcal{M}_{SL}$ to store them.
3: **for** iteration $k = 1$ to $K$ **do**
4:     Using **DRL**, find the best response $\boldsymbol{\pi}_k^* := \boldsymbol{\pi}^{\theta_k^*}$ such that:

$$\boldsymbol{\pi}_k^* = \arg\max_{\boldsymbol{\pi}} J_{\mu_0, \bar{\boldsymbol{\pi}}_{k-1}}^N(\boldsymbol{\pi})$$

5:     Collect $N_{sa}$ state-action samples of the form $(t, s, a)$ using $\boldsymbol{\pi}_k^*$ and store in $\mathcal{M}_{SL}$.
6:     Train the **NN policy** $\bar{\boldsymbol{\pi}}_k := \bar{\boldsymbol{\pi}}^{\bar{\theta}_k}$ using supervised learning to minimize the categorical loss:

$$\mathcal{L}_{\text{NLL}}(\bar{\theta}) = \mathbb{E}_{(t,s,a) \sim \mathcal{M}_{SL}} \left[ -\log \bar{\boldsymbol{\pi}}^{\bar{\theta}}(a|t,s) \right]$$

    This NN aims to mimic the behaviour of the average policy $\frac{1}{k}(\boldsymbol{\pi}_0^* + \cdots + \boldsymbol{\pi}_k^*)$.
7: **end for**
8: **return** $\bar{\boldsymbol{\pi}}^{\bar{\theta}_K}$

---

### C.1 DEDA-FP components

In **DEDA-FP** (Algo. 3), both the overall orchestration and the individual components are chosen for a specific purpose. While other choices could be made, we explain below the rationale behind our choices.

**Fictitious Play:** this is the backbone of our method. The main advantage is that it is known to converge in larges classes of games. One drawback is that convergence can be lower than some other methods, but we preferred to sacrifice the convergence speed and ensure robustness rather than the opposite.

**DRL for Best Response:** Policies are functions defined on the continuous state and action spaces so they are infinite dimentsional. Hence we had to approximate them using parameterized functions. We chose neural networks due to the empirical success in a variety of machine learning tasks. As for the training, model-free RL has the advantage to avoid exact dynamic programming and hence scale well to highly complex problems. In the implementation we chose SAC and PPO but other choices could be made, depending on the specific MFG at hand.

**Supervised learning for average policy:** In general, convergence results for Fictitious Play are not for the last iterate policy (the best response computed in the last iteration) but only for the average policy. So computing the average policy is crucial to ensure convergence. However, our policies are neural networks and averaging neural networks is hard due to non-linearities. We thus have to train a new neural network for the average policy. Here, we chose to use supervised learning, drawing inspiration from Neural Fictitious Self-Play ([Heinrich and Silver, 2016]). **Conditional Normalizing Flow (CNF) for the Mean-Field:** This is a critical design choice that directly enables one of our paper's main contributions. To solve MFGs with local density dependence (e.g., congestion), we

require a model that can both (1) sample from the population distribution and (2) compute its exact probability density at any given point in time and space ($p(x|t)$). Normalizing Flows (NFs) are well-established generative models that have been introduced precisely to provide both of these capabilities without time and CNFs are an extension of NFs, which allow us to take time into account in a natural way. Other generative models, like GANs or score-based diffusion models, can sample effectively but do not allow direct density evaluation, making them unsuitable for our goal.

# D   Implementation Details

## D.1   Time Conditioned Neural Spline Flow

We employ the Neural Spline Flow (NSF) with autoregressive layers [Durkan et al., 2019] as the flow component in our time conditioned normalizing flow.

**Neural Spline Flows**   The key idea in NSF is to transform a simple distribution (like a standard Gaussian) into a complex one using a series of invertible transformations. To make these transformations very flexible and efficient, NSF uses "rational-quadratic splines."

**Rational-Quadratic Splines**   A spline can be seen as a flexible curve made up of pieces. In our case, each piece is a "rational-quadratic" function, which is a ratio of two quadratic polynomials. These functions are smooth and can be easily inverted, which is important for our model. A rational-quadratic spline is defined by a set of $K+1$ knots $\{(x^{(k)}, y^{(k)})\}_{k=0}^{K}$. The value of the spline at a given $x$ is determined by which interval $[x^{(k)}, x^{(k+1)}]$ it falls into. Letting $\xi = (x - x^{(k)})/(x^{(k+1)} - x^{(k)})$ represent the normalized position within that interval, the spline segment is:

$$g(x) = \frac{\alpha^{(k)}(\xi)}{\beta^{(k)}(\xi)}$$

where

$$\alpha^{(k)}(\xi) = s^{(k)} y^{(k+1)} \xi^2 + [y^{(k)} \delta^{(k+1)} + y^{(k+1)} \delta^{(k)}] \xi (1 - \xi) + s^{(k)} y^{(k)} (1 - \xi)^2$$

$$\beta^{(k)}(\xi) = s^{(k)} \xi^2 + [\delta^{(k+1)} + \delta^{(k)}] \xi (1 - \xi) + s^{(k)} (1 - \xi)^2$$

and $s^{(k)} = (y^{(k+1)} - y^{(k)})/(x^{(k+1)} - x^{(k)})$ is the slope of the line connecting the knots at the interval's boundaries. $\delta^{(k)}$ represents the derivative of the spline at knot $k$.

**Autoregressive Neural Spline Flows**   In our implementation, we use the variant of NSF with autoregressive layers. This means that the parameters of the rational-quadratic spline transformation for each dimension of the data are predicted by an autoregressive neural network. Specifically, for each dimension $i$ of the input $x$, the spline parameters are computed as a function of the previous dimensions $x_{1:i-1}$:

$$\theta_i = \text{NN}(x_{1:i-1})$$

where $\text{NN}_{\text{AR}}$ denotes an autoregressive neural network. This autoregressive approach allows the model to capture complex dependencies between the dimensions of the data, as the transformation applied to each dimension is conditioned on the values of the preceding dimensions.

**Time Conditioning**   To handle the non-stationary nature of the mean-field distribution, we explicitly condition the Neural Spline Flow on time $t$. This means that the entire transformation, and specifically the parameters of the rational-quadratic splines, are made dependent on the current time step. In our autoregressive setup, the neural network that predicts the spline parameters (NN in the equation above) not only takes the previous dimensions $x_{1:i-1}$ as input but also the time $t$. The time variable $t$ is typically concatenated with the input features or fed into the neural network as an additional input, allowing the network to learn time-dependent transformations. This enables the flow to dynamically

adjust its shape and density characteristics as time evolves from $t = 0$ to $t = T$, thereby capturing the non-stationary dynamics of the mean-field.

# E    Numerical Experiments details

This section provides further experimental results and detailed comparisons between our proposed DEDA-FP approach and the benchmark algorithms considered in the main paper.

## E.1    Beach Bar Problem

Further numerical results for the Beach Bar problem are shown in Figures 8 and 9.

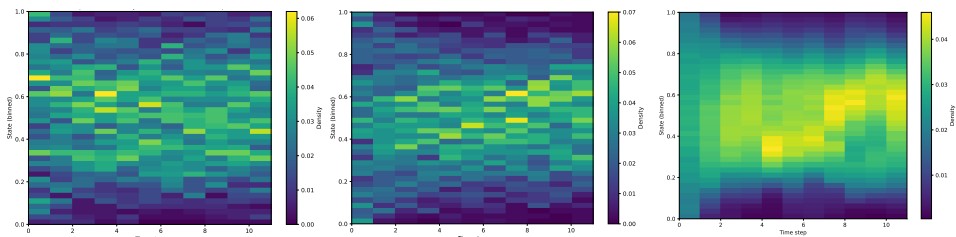

Figure 8: **Comparison of Nash equilibrium distribution heatmaps in the Beach Bar Problem**. From left to right: Algo. 1, Algo. 2, and DEDA-FP (Algo. 3). Thanks to its remarkable speed, DEDA-FP can utilize a high volume of samples (6x times in the displayed figure) for robust distribution approximation, a scale that proves computationally prohibitive for existing benchmarks.

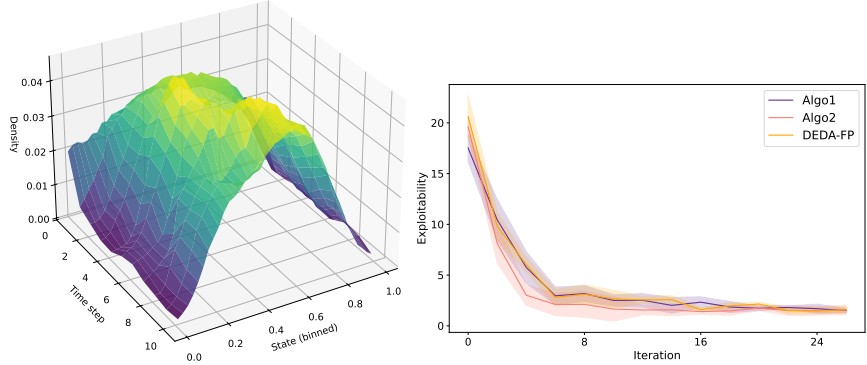

Figure 9: **Beach Bar Problem Results for DEDA-FP**. Left: Nash equilibrium distribution. Right: Exploitability decay comparison across algorithms.

## E.2    LQ model

Further numerical results for the LQ problem are shown in Figures 10 and 11.

## E.3    4-rooms exploration

Further numerical results for the 4-rooms exploration problem are shown in Figures 16 and 17.

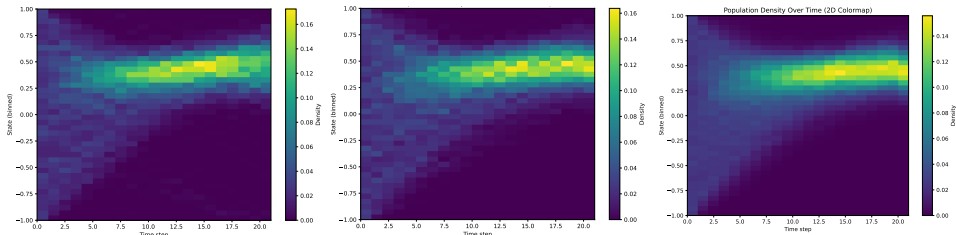

Figure 10: **Comparison of Nash equilibrium distribution heatmaps in the LQ Problem**. From left to right: Algo Algo. 1, Algo Algo. 2, and DEDA-FP (Algo. 3).

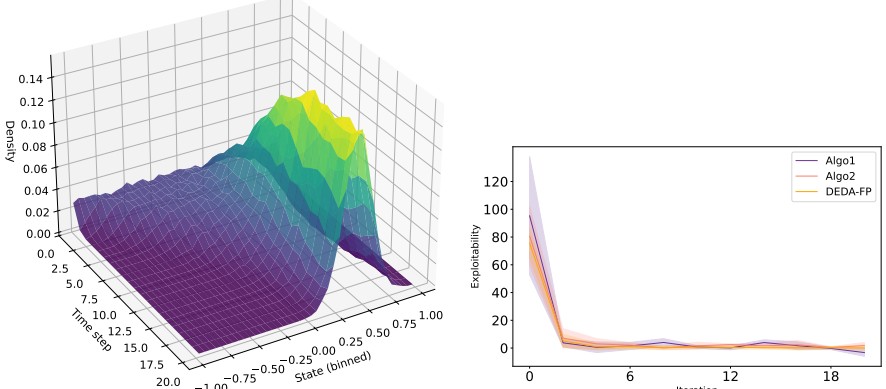

Figure 11: **LQ Problem Results for DEDA-FP**. Left: Nash equilibrium distribution. Right: Exploitability decay comparison across algorithms.

### E.4 Market Model

Here we present one more example on an environment of a different type, with a financial application. It is a discrete time version of the price impact model in MFG literature, introduced by Carmona and Lacker.[3]

**Environment:** We consider a market model where $\mathcal{X} = [-5, 5]$ represents the inventory for a stock. The action space $\mathcal{A} = [-1, 1]$ represents the rate of trading for the stock. Each agent controls the inventory for the stock. The dynamics is: $x_{t+1} = x_t + a_t + \epsilon_t$, where $\epsilon_t \sim \mathcal{N}(0, 1)$. The reward is $r(x, a, \bar{a}) = -C_{\mathcal{X}} x^2 - C_{\mathcal{A}} a^2 + hx\bar{a}$, where $C_{\mathcal{X}}$, $C_{\mathcal{A}}$, and $h$ are positive constants, $\bar{a}$ is the mean of the action. At each time $t$, the representative agent wants to minimize the shares held. In this model, the agent interacts with the distribution of the action instead of the population distribution. The mean field term $hx\bar{a}$ reflects the impact of the action on the price.

**Numerical results:** Results are shown in Figures 12, 13 and 14. We observe that traders tend to liquidate their portfolios (given to the $x^2$ term in the reward function). However, a proportion of agents is incentivized to buy instead of sell due to the interaction term. Moreover, we observe that our model (DEDA-FP) consistently provides a superior representation of the distribution, which is ensured by its efficiency in sampling a large number of agent positions at every time step.

---

[3]René Carmona and Daniel Lacker. A probabilistic weak formulation of mean field games and applications. *Annals of applied probability: an official journal of the Institute of Mathematical Statistics*, 25(3):1189–1231, 2015.

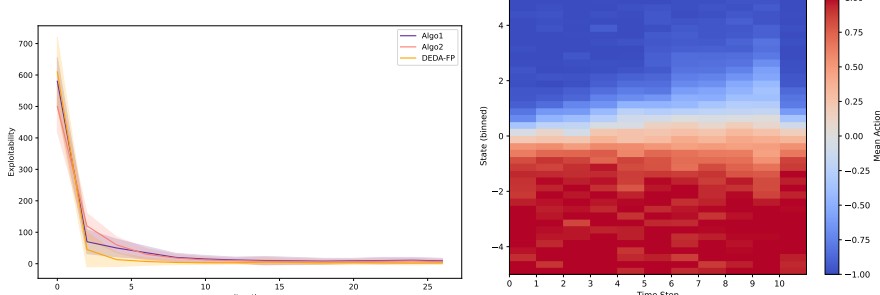

Figure 12: **Market Model Results:** Left: Exploitability decay comparison across algorithms; Right: Nash Equilibrium Policy learned by DEDA-FP

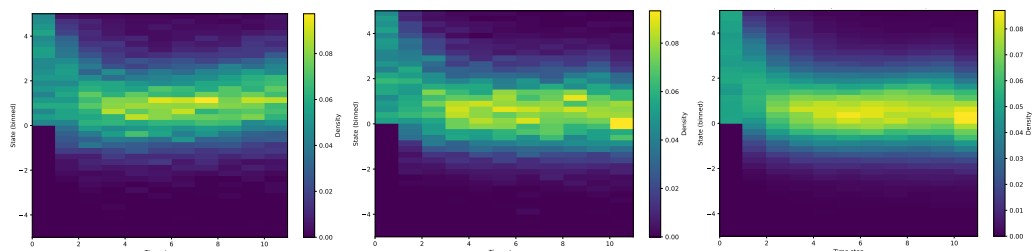

Figure 13: **Comparison of Nash equilibrium distribution heatmaps in the Market Model Problem**. From left to right: Algo Algo. 1, Algo Algo. 2, and DEDA-FP (Algo. 3).

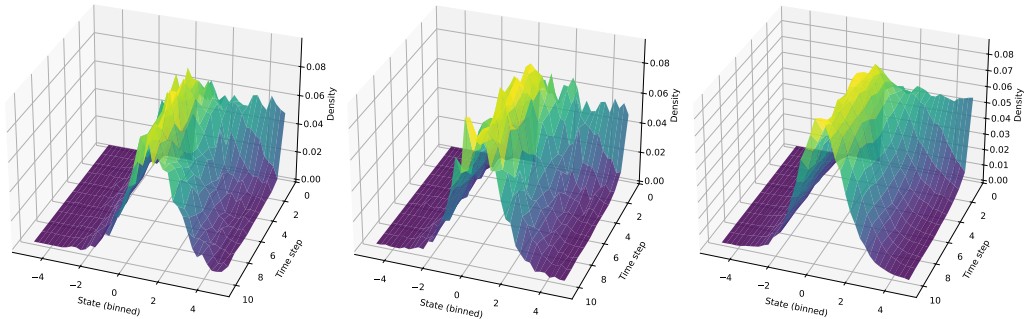

Figure 14: **Comparison of Nash equilibrium distribution in the Market Model Problem ((3D plots))**. From left to right: Algo Algo. 1, Algo Algo. 2, and DEDA-FP (Algo. 3).

# F   Hyperparameter Sweep

We sweep the learning rate over the set $\{3 \times 10^{-2}, 3 \times 10^{-3}, 3 \times 10^{-4}, 3 \times 10^{-5}, 3 \times 10^{-6}\}$ for Deep RL in Algo. 1 in to the center environment shown in Figure 15. We observe that a learning rate of $3 \times 10^{-4}$ yields more stable training and faster convergence. Based on this observation, we adopt $3 \times 10^{-4}$ for the Deep RL component in Algo. 2 and Algo. 3 as well.

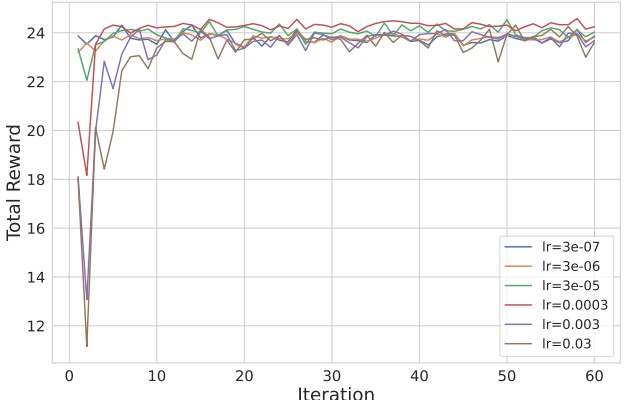

Figure 15: Total reward vs iterations for different learning rate of Deep RL in Algo. 1 in the center environment

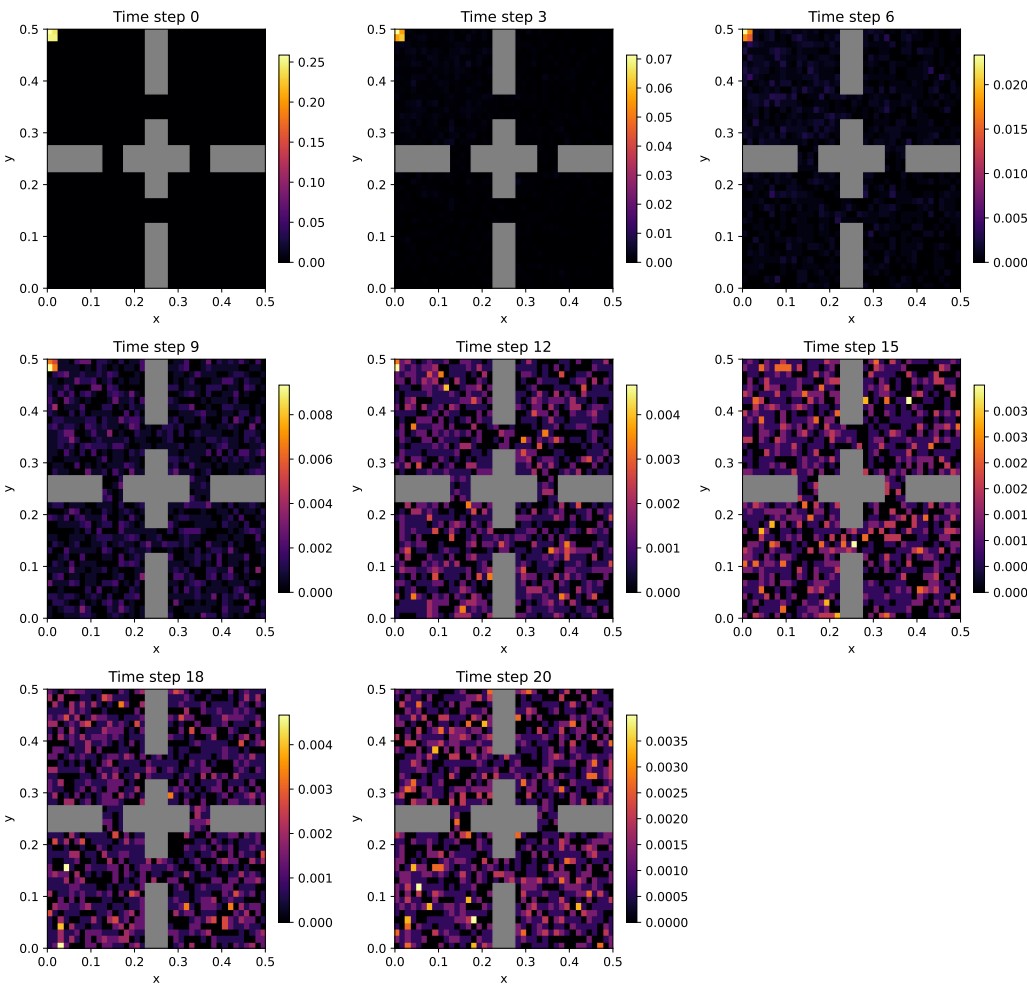

Figure 16: **4 rooms explorations**. Nash Equilibrium mean field flow obtained by Algo. 2 sampling 1500 trajectories

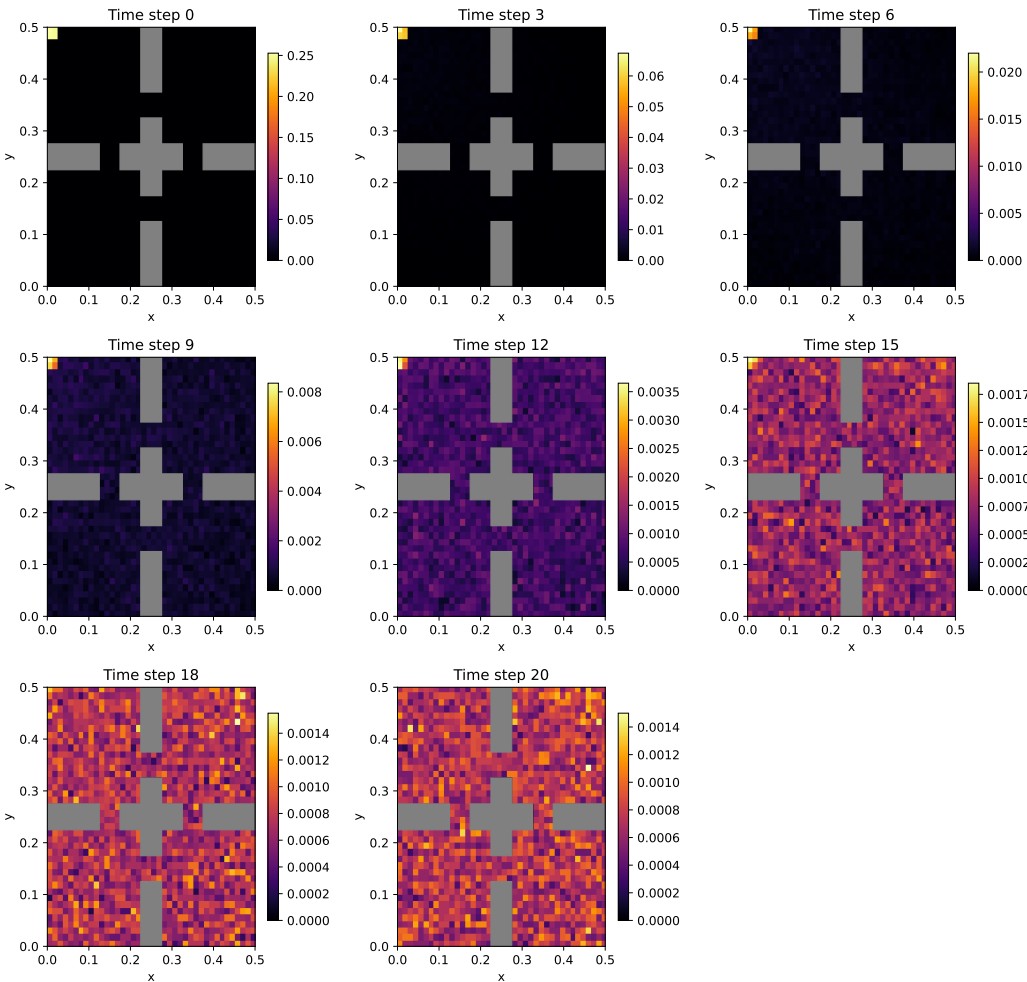

Figure 17: **4 rooms explorations**. Nash Equilibrium mean field flow obtained by Algo. 3 sampling 8000 trajectories $10x$ faster than Algo. 2 and Algo. 1.

