# OpenReview forum: "Solving Continuous Mean Field Games: Deep Reinforcement Learning for Non-Stationary Dynamics"
_NeurIPS.cc/2025/Conference — NeurIPS 2025 poster_

### Official Review · Reviewer_ZLz3 · 2025-07-02

**Clarity:** 3
**Significance:** 2
**Originality:** 3
**Rating:** 5
**Confidence:** 2

**Summary:**

This paper proposes Density-Enhanced Deep-Average Fictitious Play (DEDA-FP), a novel DRL algorithm for non-stationary Mean Field Games (MFGs) with continuous state-action spaces. DEDA-FP extends Fictitious Play by combining DRL for best responses, supervised learning for average policy, and a time-conditioned Conditional Normalizing Flow for dynamic population modeling. Experiments (Beach Bar, Linear-Quadratic, 4-rooms) show DEDA-FP handles local dependencies, achieves 10× faster sampling, and converges to Nash equilibria.

**Questions:**

1. The average policy is obtained through three layers of approximation: training error from the NLL loss, sampling error due to limited samples at each time step, and learning error when estimating the best response. To what extent do these errors impact the final average policy?

2. Is it theoretically sound to use an average policy in a time-varying MFG? Since the theoretical foundation of NFSP is established for time-independent settings, does this assumption still hold in a non-stationary context?

**Ethical Concerns:**

["NO or VERY MINOR ethics concerns only"]

**Final Justification:**

The author had an active discussion and also answered my questions. Overall, this paper has achieved good results in solving time-varying continuous MFGs. Therefore, I suggest accepting this paper.

**Limitations:**

yes

**Quality:**

3

**Strengths And Weaknesses:**

1. DEDA-FP is the first method to tackle non-stationary MFGs with continuous state and action spaces, overcoming the limitations of previous approaches that focused on finite or stationary models.

2. It combines DRL, supervised learning, and Conditional Normalizing Flow in an innovative and effective way to learn optimal policies while tracking the evolving population dynamics.

3. As noted under the “limitation” section, the paper does not provide a theoretical guarantee for the convergence of DEDA-FP, such as the conditions under which the algorithm converges to a Nash equilibrium. This is a key shortcoming, as theoretical rigor is important to demonstrate the method’s stability.

4. Limited generalization testing: Although the three experiments increase in complexity, they may still be overly simplified compared to real-world scenarios.

---

> ### Author Rebuttal · Authors · 2025-07-30
>
> First of all, we would like to sincerely thank the reviewer for their detailed review, which highlights our contributions and originality (*"first method to tackle non-stationary MFGs with continuous state and action spaces"*), and for the constructive suggestions to improve our manuscript. As suggested, we have added a theoretical discussion to the manuscript concerning the convergence of DEDA-FP.
>
> **W1 (Theoretical guarantees):**
> Please see the detailed answer to **Q1 (Error analysis)** below. We are happy to provide more details if needed.
>
> **W2 (Real world scenarios):**
> We agree that real-world scenarios are the ultimate target and this is the very reason why we develop an algorithm for continuous space MFGs instead of discrete space (or LQ) as a the rest of the literature. The examples we solve are of comparable complexity to those presented in other well-established papers such as: [Guo et al., 2019, Subramanian and Mahajan, 2019, Elie et al., 2020, Fu et al., 2019, Cui and Koeppl, 2022, Angiuli et al., 2022] cited in our introduction, as well as other recent works on RL for MFGs, such as: [Zhou et al., ] (ICML'24) which studies a simple one-dimensional model, [Zhang et al., 2024] (ICLR'25) which studies only a finite-state MFGs with very few states,  or [Ocello et al., 2025] (ICML'25) which contains a 2-dimensional example similar to our Example 3 except that they only consider a finite state version while we solve a continuous space model (which is muc more realistic for crowd motion modeling).
>
> **Q1 (Error analysis):** Thank you for raising this interesting point. Although a complete analysis is beyond the scope of our paper, we explain below that there is a clear path to analyze the different sources of errors and how they compound to yield an approximate Nash equilibrium. (We will include the following discussion in the revised paper.)
>
> ---
> *On the convergence of DEDA-FP:*
>
> While a full theoretical analysis is left for future work, we can provide strong arguments for the convergence of DEDA-FP, based upon existing theory.
>
> At each iteration, there are **three sources of errors**: **(e1)** best response computation (using DRL), **(e2)** average policy computation (supervised learning), and **(e3)** mean-field approximation (with CNF). One can expect convergence towards an **approximate** mean-field Nash equilibrium, i.e., an $\epsilon-$Nash equilibrium, where $\epsilon$ depends on the errors. A convergence proof can be carried out by analyzing *how the errors propagate* through iterations.
>
> While most of the literature on FP for MFG focuses on perfect computations, the work by [Elie et al., 2020] have carried out such an error propagation for FP when there is only the first type of error, namely, on the BR computation.
> We argue that a similar proof technique can be used to track the impact of other errors on the final approximate equilibrium.
> In this direction, we now outline why these three types of errors can be made arbitrarily small:
>
> - **(e1) Best Response Error:** This error comes from the DRL algorithm. We expect this to be the most serious source of errors because solving an MDP is more difficult than a supervised learning task or a distribution-approximation task. Fortunately, this is precisely the type of error analyzed in [Elie et al. 2020], which have shown that it does not prevent convergence to an approximate equilibrium. Furthermore, to make sure that the inner loop of DRL computes a very good policy, we have carried out sweeps over hyperparameters (see Fig. 14 in Appendix for an example). This guarantees that our choice of hyperparameters yields a small error for **(e1)**.
>
> - **(e2) Average Policy Error:** This is a (relatively standard) supervised learning task. Thanks to the universal approximation properties of neural networks, the error in representing the true average of past policies can be made arbitrarily small provided the neural network is sufficiently deep/large, the training set is sufficiently large, and there are sufficiently many training steps.
>
> - **(e3) Distribution Error:** The specific architecture we use, Neural Spline Flows, has been shown to be a universal distribution approximator [Durkan, 2019]. This guarantees that the error **(e3)** can be made arbitrarily small if the model is expressive enough.
>
> ---
>
> **Q2 (average policy in time dependent case):**
>
> Thank you for raising this interesting question. We address the two points:
>
> 1. Yes, it is sound to use an average policy in time-varying MFGs, as has been shown in [Cardaliaguet and Hadikhanloo, 2017] and [Perrin et al., 2020], which both analyzed fictitious play in finite-horizon MFGs. However, there is a twist, which is that the weights depend on the population distribution, as shown in the formula at the bottom of page 3 in [Perrin et al., 2020]. This is the proper definition of average policy in time-dependent MFGs.
>
> 2. You are right that we do not know any theoretical foundations for NFSP in time-dependent games. However, we are not using exactly NFSP. What we really use is the idea of supervised learning for the average policy, and this part is sound because we make sure that the policy learns the time dependence on top of the state-dependence.
>
> It is important to notice that the way we build the replay buffer of state-action pairs ensures two key properties: **(1)** we store an equal number of samples at each iteration, which implies that sampling uniformly from the replay buffer amount to sample uniformly from the past iterations; and **(2)** the samples are generated by using the current best response, which implies that the probability to see a given state-action pair is directly linked to the mean field generated by the best response (this is what is required in the aforementioned formula of [Perrin et al., 2020]).
>
>
> **We hope that this addresses the reviewers' questions about the theoretical foundation of our DEDA-FP algorithm and that the reviewer will consider raising their score. Otherwise, we are happy to discuss these points further.**
>
>
> *References:*
>
> [Ocello et al., 2025] Ocello, Antonio, et al. "Finite-Sample Convergence Bounds for Trust Region Policy Optimization in Mean Field Games." Forty-second International Conference on Machine Learning.
>
> [Zhou et al., 2024] Zhou, Fuzhong, et al. "Graphon Mean Field Games with a Representative Player: Analysis and Learning Algorithm." International Conference on Machine Learning. PMLR, 2024.
>
> [Elie et al., 2020] Elie, Romuald, et al. "On the convergence of model free learning in mean field games." Proceedings of the AAAI Conference on Artificial Intelligence. Vol. 34. No. 05. 2020.
>
> [Durkan, 2019] Durkan, Conor, et al. "Neural spline flows." Advances in neural information processing systems 32 (2019).
>
> [Zhang et al., 2024] Zhang, Fengzhuo, et al. "Learning regularized graphon mean-field games with unknown graphons." Journal of Machine Learning Research 25.372 (2024): 1-95.
>
> [Cardaliaguet and Hadikhanloo, 2017] Cardaliaguet, Pierre, and Saeed Hadikhanloo. "Learning in mean field games: the fictitious play." ESAIM: Control, Optimisation and Calculus of Variations 23.2 (2017): 569-591.
>
> [Perrin et al., 2020] Perrin, Sarah, et al. "Fictitious play for mean field games: Continuous time analysis and applications." Advances in neural information processing systems 33 (2020): 13199-13213.

---

> > ### Comment · Reviewer_ZLz3 · 2025-08-02
> >
> > Thank you for the author's reply. It seems that theoretical guarantees remain a weakness. For this type of heuristic algorithm, I hope to see sufficiently extensive and complex experiments to prove its feasibility and reliability. However, although the current experimental section has a certain degree of complexity and the proposed algorithm can solve these scenarios well, it may still be insufficient. These two weaknesses make it unknown whether the proposed algorithm can solve more complex and realistic time-varying continuous MFGs. However, this work still provides some references and inspirations for future work in this direction, which is why I gave a borderline accept.
> >
> > In addition, I have some questions:
> >
> > 1. What is the specific definition of $J$ in line 4 of Algo. 3?
> > 2. In time-varying MFGs, would the average policy with uniform weights be the best? Would giving more weights to the newer best response policy lead to better results?

---

> > > ### Author Response · Authors · 2025-08-05
> > > **Proof of convergence of DEDA-FP**
> > >
> > > Due to space constraints the previous answer did not sufficiently address the theoretical gap raised by the reviewer. In response to this concern, we are glad to provide  a proof of the convergence for DEDA-FP. **We are confident that this theoretical analysis complements our strong experimental findings and significantly strengthens the contribution of our paper**. We plan to include it in the camera-ready version.
> > >
> > >
> > > **Main theorem (convergence to approximate Nash equilibrium).** Under suitable assumptions (see below), we have the bound:
> > > $$
> > >   e_k^{true} \le C_0 \epsilon_{cnf}^k + \frac{e_0}{k} + \frac{1}{k}\sum_{i=1}^k \left( (i+1)\epsilon_{br}^{i+1} + C_1(\epsilon_{sl}^{i+1} + \epsilon_{cnf}^{i+1}) + \frac{C_2}{i} \right)$$
> > >   for some constants $C_0, C_1, C_2 > 0$, where $e_k^{true} := J(BR(\mu^{\bar\pi_k}), \mu^{\bar\pi_k}) - J(\bar\pi_k, \mu^{\bar\pi_k}) \ge 0$ is the **true exploitability**, which measures the incentive to deviate from the policy $\bar\pi_k$ in the true distribution it generates, $\mu^{\bar\pi_k}$.
> > >
> > > -----
> > > We first formalize the **error sources**. We denote by $d_1$ the time-extended Wasserstein distance:
> > > $d_1(\mu_1,\mu_2) = \max_{t=0,\dots,N_T} W_1(\mu_{1,t}, \mu_{2,t})$ where $W_1$ is the Wasserstein-1 distance on $\mathcal{X}=\mathbb{R}^d$.
> > >
> > > 1. **Best Response Error.** The sub-optimality of the DRL policy $ \pi_k^* $ wrt the mean-field flow
> > > $\mathcal G_{k-1}$ from the previous iteration: $\epsilon_{br}^k := J(BR(\mathcal G_{k-1}), \mathcal G_{k-1}) - J(\pi_k^*, \mathcal G_{k-1}) \ge 0$.
> > >
> > > 2. **Average Policy Error.** The error in the supervised learning step: $\epsilon_{sl}^k:= d(\bar\pi_k, \Pi_k^{true})$ where $\Pi_k^{true} = \frac{1}{k}\sum_{i=1}^k \pi_i^*$ is the true average policy.
> > >
> > > 3. **Distribution Error.** The error of the CNF model $\mathcal G_k$ in approximating the true mean-field flow $\mu^{\bar\pi_k}$:  $\epsilon_{cnf}^k:=d_1(\mathcal G_k, \mu^{\bar\pi_k})$.
> > >
> > > We will use the following two assumptions, which are satisfied under mild conditions on the reward and transition functions:
> > >
> > > **Assumption 1** (Lipschitz Continuity of $J$). For any policy $\pi$ and any two flows $\mu_1, \mu_2$: $|J(\pi, \mu_1) - J(\pi, \mu_2)| \le L \cdot d_1(\mu_1, \mu_2)$.
> > >
> > > **Assumption 2** (Lipchitz continuity of MF). For any policies $\pi_1,\pi_2$, the generated mean fields satisfy: $d_1(\mu^{\pi_1}, \mu^{\pi_2}) \le L_{mf} d(\pi_1,\pi_2)$.
> > >
> > > We first bound the exploitability with respect to the tractable distribution $\mathcal G_k$ (**Lemma 1**). Then we connect this result with the true value of the exploitability to show that the pair $(\bar\pi_k, \mu^{\bar\pi_k})$ is an approximate Nash equilibrium (**Lemma 2**). Combining these two bounds yield the **Main theorem** stated above.
> > >
> > > ---
> > > **Lemma 1.** Let Assumption 1 from Elie et al. (2020) and Assumptions 1 and 2 above hold. The tractable exploitability $e_k$ satisfies the following bound:
> > > $$e_k \le \frac{e_0}{k} + \frac{1}{k}\sum_{i=1}^k \left( (i+1)\epsilon_{br}^{i+1} + C_1(\epsilon_{sl}^{i+1} + \epsilon_{cnf}^{i+1}) + \frac{C_2}{i} \right)$$ for some constants $C_1, C_2 > 0$, where $e_k := J(BR(\mathcal G_k), \mathcal G_k) - J(\bar\pi_k, \mathcal G_k)$ is the **tractable exploitability** defined with respect to the learned distribution $\mathcal G_k$.
> > >
> > > *Proof Sketch*.
> > >
> > > The proof adapts the derivation of estimate (8) in Theorem 6 of Elie et al. (2020) by analyzing the recursive term $(k+1) \hat{e}_{k+1} - k\hat{e}_k$, where  $\hat{e}_k$ is the approximate exploitability.
> > >
> > > The crucial step is bounding the change in the value function, $|J_{k+1} - J_k|$, which is bounded by the distance between successive learned distributions, $d_1(\mathcal{G}_{k+1}, \mathcal{G}_k)$.
> > >
> > > We then bound this distance using the triangle inequality, which introduces terms dependent on our defined errors: the CNF error ($\epsilon_{cnf}$), the supervised learning error ($\epsilon_{sl}$), and the inherent $O(1/k)$ stability of the exact Fictitious Play update (from Lemma 5 in Elie et al.). Substituting these bounds into the recursion introduces additive terms for each error source. Solving this final inequality using Lemma 9 from the reference paper yields the stated theorem. $\square$
> > >
> > > ---
> > > **Lemma 2.** Under the same assumptions, the true exploitability $e_k^{true}$ is close to the tractable exploitability $e_k$. Specifically, $$ |e_k^{true} - e_k| \le 4L \cdot \epsilon_{cnf}^k$$ where $L$ is the Lipschitz constant of Assumption 1.
> > >
> > > *Proof sketch.* We bound the difference between the terms of $e_k$ and $e_k^{true}$ using Assumption 1. We have:  $$|J(\bar\pi_k, \mathcal G_k) - J(\bar\pi_k, \mu^{\bar\pi_k})|\le L \cdot d_1(\mathcal G_k, \mu^{\bar\pi_k}) \le L \cdot \epsilon_{cnf}^k $$
> > > A similar bound holds for $|J(BR(\mathcal G_k), \mathcal G_k) - J(BR(\mu^{\bar\pi_k}), \mu^{\bar\pi_k})|$, by using the stability of the best-response map under the same assumptions. Combining these arguments using triangule inequality we conclude. $\square$

---

> > > > ### Author Response · Authors · 2025-08-05
> > > > **Response to Questions on Algorithm and Averaging Method**
> > > >
> > > > **QUESTIONS**
> > > >
> > > > **Q1:** The approximate total expected reward in Algo 3 (line $4$), namely $J^N_{\mu_0, \bar\pi_{k-1}, \bar{G}_{k-1}}$, is defined as:
> > > >
> > > > $J^N_{\mu_0, \bar\pi_{k-1}, \bar G_{k-1}}(\pi)= \sum_{t=0}^{N_T}\left( \frac{1}{N}  \sum_{i=1}^{N}r( x_t^{(i)}, a_t^{(i)}, \bar{G}_t^{(k-1)}) \right)$
> > > >
> > > > where for each sample $i \in \{1, \dots N\}$, $x_0^{(i)}\sim\mu_0$ , $a_t^{(i)}\sim \pi (\cdot | x_t^{(i)})$ and  $x_{t+1}^{(i)}$ follows the game dynamics. The population distribution at time $t$ is described by $\bar G_t^{(k-1)}$ (learned using trajectories sampled from $\bar\pi_{k-1}$ ).
> > > >
> > > >
> > > > ---
> > > > **Q2** **Uniform vs. Weighted Averaging of Policies**.
> > > > We use uniform weighting because standard Fictitious Play (FP) has convergence guarantees, while weighted versions do not, to our knowledge. This is crucial because Nash Equilibria often require mixed (stochastic) strategies. Uniform averaging over past best responses allows the algorithm to learn this necessary mixture over time.
> > > >
> > > > In contrast, giving more weight to recent best responses, which are often less stochastic, would bias the policy, preventing convergence to the true mixed equilibrium.
> > > >
> > > > ----
> > > >
> > > > We hope these clarifications regarding the theoretical guarantees (previous comment) and the average policy have fully addressed your concerns.  We hope you will consider raising your score, and we welcome the opportunity to discuss these points further.

---

> > > > > ### Comment · Reviewer_ZLz3 · 2025-08-05
> > > > >
> > > > > I'm glad that the authors can provide a convergence proof to enhance the theoretical nature and contributions of the article, and I will increase my score.

---

### Official Review · Reviewer_uqaE · 2025-07-02

**Clarity:** 3
**Significance:** 2
**Originality:** 2
**Rating:** 5
**Confidence:** 3

**Summary:**

This paper introduces a novel deep RL algorithm DEDA-FP for addressing non-stationary Mean Field Games that have continuous state and action spaces. Their approach is based on fictitious play from classical game theory, actor-critic or PPO from DRL, and the use of supervised learning over the policies across iterations of fictitious play.  This combination allows the authors to compute the Nash equilibrium policy and mean field distribution. They demonstrate their methodology in three different experimental settings, showing that DEDA-FP can indeed handle learning in non-stationary MFGs with continuous action spaces.

**Questions:**

The authors clearly indicate the advantages of the generality of DEDA-FP along the dimensions in Table 1, but are there any dimensions that DEDA-FP cannot cover compared to any of these previous works? Or any drawbacks? Understanding weaknesses compared to other approaches as well as the stated advantages could help exactly clarify the contribution of this work.

Can DEDA-FP be applied mostly out of the box to MFGs that are discrete? Or would there be no reason to apply it in such settings? Would we expect worse performance compared to some other approaches if there are discrete action spaces or could  DEDA-FP be the algorithm of choice even in such settings?

Combining fictitious play, DRL, supervised learning. and the CNF clearly gives strong numerical results, but could the authors provide some rationale on why these methods in specific? Would swapping out one of these components for something else still be able to give comparable results or is each part clearly better than alternatives for some reason(s)? Is the orchestration more important than the individual components or is each individually crucial? I feel the lack of theoretical discussion behind why these combination of parts work well is part of what begs these questions.

**Ethical Concerns:**

["NO or VERY MINOR ethics concerns only"]

**Final Justification:**

I thank the authors again for addressing the concerns stated in my original review. Additionally, it is clear that the authors were able to provide more substantial theoretical justification for DEDA-FP. The discussion with Reviewer ZLz3 was informative and certainly strengthens the depth of the contribution. Mainly considering this factor on top of my already weakly positive assessment and the authors' responses to my initial questions, I have decided to raise my score to a 5.

**Limitations:**

Yes.

**Paper Formatting Concerns:**

None noticed.

**Quality:**

3

**Strengths And Weaknesses:**

The authors clearly highlight their contributions in terms of which classes of MFGs they are able to handle, including non-stationary ones and ones with continuous action spaces. This is augmented with Table 1 that draws clear comparisons to the literature on the covered classes of MFGs. The advantages of the generality of DEDA-FP along these dimensions compared to prior works is evident. The exposition in section 3 nicely builds up to DEDA-FP first detailing a simple version of fictitious play and then learning of the NE policy while detailing what is missing along the way to presenting DEDA-FP. This helps the reader understand the challenges addressed by DEDA-FP. The numerical experiments are very clearly described and well presented visually, and again build progressively demonstrating increased challenges of the considered settings. The ability of DEDA-FP to address settings not captured by other algorithms is well presented and compelling.

Although this paper is primarily supported by its experimental results, perhaps some more theoretical discussion around DEDA-FP would strengthen the work. It is not clear the significance of each individual part of the pipeline of DEDA-FP. In terms of which DRL alg is used, which supervised alg is used, etc, how much of the approach is based in the orchestration of these methods compared to the individual strengths of each component? The authors do clearly mention themselves in limitations that it is difficult to obtain a theoretical understanding due to the complexity of analyzing deep neural nets, but I do feel like there could still be sone room to help the reader understand why DEDA-FP works well.

---

> ### Author Rebuttal · Authors · 2025-07-30
>
> We sincerely thank the reviewer for the constructive suggestions to improve our paper. We are grateful for their acknowledgment that *"the advantages of the generality of DEDA-FP... compared to prior works is evident"*, *"The exposition in section 3 nicely builds up to DEDA-FP*", and that *"the numerical experiments are very clearly described and well presented visually"*.
> We provide detailed responses to each of your questions below.
>
> **Weakness (Lack of theoretical guaranteed)**:
> We thank the reviewer for their interesting question. We address this point in **Q1 (Error analysis) of Reviewer ZLz3**, where we have provided a detailed discussion on the convergence of DEDA-FP. We would be glad to engage in further discussion on this topic..
>
> **Q1 (Drawbacks compared to previous works)**:
> Thank you for this question. We will stress the limitations of our work in the revised paper.
> 1. The primary drawback of our approach does not relate to the scope of problems it can handle, but rather to the **slow convergence** of the underlying Fictitious Play (FP) algorithm in some cases.
> While other methods such as those based on entropy-regularized fixed-point iterative methods [Cui et al., 2021] or Online Mirror Descent (OMD) [Perolat et al., 2022] may converge faster, they have so far only been demonstrated in **less complex, finite state-space settings** and present their own implementation and convergence challenges. We chose FP because it offers a robust and stable framework for tackling the main challenges we set out to solve continuous spaces, non-stationary dynamics, and learning the mean-field density, prioritizing stability over convergence speed.
> 2. A second limitation is that DEDA-FP requires **more training time per FP iteration** compared to the simpler Algo 1 and Algo 2. Specifically, training the Conditional Normalizing Flow (CNF) to accurately model the time-dependent mean-field distribution adds computational overhead to each iteration. While this leads to a richer and more accurate solution, especially in problems with local density dependence, it comes at the cost of increased single-iteration training time.
>
>
> **Q2 (Discrete action space):**
> Yes, DEDA-FP can be applied to discrete spaces. This is because we can embed discrete spaces in continuous ones and then use our algorith. However this naive implementation would be highly inefficient. A tailored implementation would be far more effective than a naive one.
>
> Indeed, a naive application would require embedding the discrete space into a continuous one and projecting the outputs, which is inefficient. A proper, adapted implementation would be more powerful and would involve **two** key changes:
> 1. The *policy network* would use a softmax output layer instead of modeling a Gaussian distribution to directly handle a probability distribution over discrete actions.
> 2. The *Conditional Normalizing Flow* (CNF), designed for continuous densities, would be replaced by a model suited for distributions on discrete state spaces. Here again, one could use a neural network with a sofmax on the last layer.
>
> **Q3 (Components of the algo):**
> Thank you for giving us the opportunity to justify our choices. Both the overall orchestration and the individual components are chosen for a specific purpose. While other choices could be made, we explain below the rationale behind our choices.
>
> - **Fictitious Play:** this is the backbone of our method. The main advantage is that it is known to converge in larges classes of games. One drawback is that convergence can be lower than some other methods, but we prefered to sacrifice the convergence speed and ensure robustness rather than the opposite.
>
> - **DRL for Best Response:** Policies are functions defined on the continuous state and action spaces so they are infinite dimentsional. Hence we had to approximate them using parameterized functions. We chose neural networks due to the empirical success in a variety of machine learning tasks. As for the training, model-free RL has the advantage to avoid exact dynamic programming and hence scale well to highly complex problems. In the implementation we chose SAC and PPO but other choices could be made, depending on the specific MFG at hand.
>
> - **Supervised learning for average policy:** In general, convergence results for Fictitious Play are not for the last iterate policy (the best response computed in the last iteration) but only for the average policy. So computing the average policy is crucial to ensure convergence. However, our policies are neural networks and averaging neural networks is hard due to non-linearities. We thus have to train a new neural network for the average policy. Here, we chose to use supervised learning, drawing inspiration from Neural Fictitious Self-Play [Heinrich et al., 2016]. To the best of our knowledge,  this is a state-of-the-art method but we are open to suggestions.
>
> - **Conditional Normalizing Flow (CNF)** for the Mean-Field: This is a critical design choice that directly enables one of our paper's main contributions. To solve MFGs with local density dependence (e.g., congestion), we require a model that can both (1) **sample** from the population distribution and (2) compute its exact probability **density** at any given point in time and space ($p(x|t)$). Normalizing Flows (NFs) are well-established generative models that have been introduced precisely to provide both of these capabilities without time and CNFs are an extension of NFs which allow us to take time into account in a natural way. Other generative models, like GANs or score-based diffusion models, can sample effectively but do not allow direct density evaluation, making them unsuitable for our goal.
>
> **We will include a discussion along these lines in the main text if space permits, or in the appendix of the final paper.
> We hope these arguments, especially on the theoretical foundations of DEDA-FP, clarify our approach and that the reviewer will consider raising their score. Of course, we remain open to further discussion on these points.**
>
>
>
> *References:*
>
> [Cui et al., 2021] Cui, Kai, and Heinz Koeppl. "Approximately solving mean field games via entropy-regularized deep reinforcement learning." International Conference on Artificial Intelligence and Statistics. PMLR, 2021.
>
> [Perolat et al., 2021] Pérolat, Julien, et al. "Scaling Mean Field Games by Online Mirror Descent." Proceedings of the 21st International Conference on Autonomous Agents and Multiagent Systems. 2022.
>
> [Heinrich et al., 2016] Heinrich, Johannes, and David Silver. "Deep reinforcement learning from self-play in imperfect-information games." arXiv preprint arXiv:1603.01121 (2016).

---

> ### Comment · Reviewer_uqaE · 2025-08-04
>
> I appreciate the thorough response of the authors, in particular the discussion on the convergence of DEDA-FP. I also appreciate the expanded rationale on the choices in the design of the algo. I feel more confident in my previous assessment and will maintain it as is. Thank you again.

---

> > ### Author Response · Authors · 2025-08-06
> >
> > We would like to thank you for the time you took to read our answers and for your reply. We also take this opportunity to stress that, in response to a question of **reviewer ZLz3**, we have provided a detailed sketch of the **analysis of error propagation**.
> >
> > We hope that this will convince you to raise your score to an **accept** but if you have any further question, we will try our best to reply before the author-reviewer discussion phase ends. Thank you again for your time and attention.

---

### Official Review · Reviewer_51K9 · 2025-07-11

**Clarity:** 3
**Significance:** 3
**Originality:** 2
**Rating:** 5
**Confidence:** 3

**Summary:**

This is an experimental paper that focuses on studying deep RL methods for solving non-stationary MFGs with continuous state/action spaces. Using standard deep RL and a neural network approach, the Nash equilibrium policy can be learnt. However, there are no known approaches in the literature for how to approximate the equilibrium mean field, i.e. the sequence of population distributions at equilibrium. Given the non-stationarity of general MFG settings, this aspect is crucial. The authors propose Density-Enhanced Deep-Average Fictitious Play (DEDA-FP), a heuristic algorithm for approximating a the equilibrium mean field in these games. The algorithm leverages deep learning techniques such as a time-dependent conditional normalizing flow (CNF), which allows for sampling from the mean field distribution. DEDA-FP is evaluated empirically against standard techniques and shows significant improvements for sampling efficiency.

**Questions:**

See Weaknesses above.

**Ethical Concerns:**

["NO or VERY MINOR ethics concerns only"]

**Final Justification:**

- My concerns have been appropriately addressed, especially since the authors have provided a sketch of a convergence result for their proposed algorithm DEDA-FP.
- The authors seem committed to adding this result to a camera ready version, along with incorporating the changes suggested by other reviewers. This would make the paper a nice contribution to a challenging problem, with justification as to _why_ their approach is expected to generalize, and with ample directions for future work. As such, I will adjust my score to an accept.

**Limitations:**

Yes.

**Quality:**

3

**Strengths And Weaknesses:**

**Strengths:**

The paper is clear in its exposition and nicely builds up to the introduction of DEDA-FP. There is no doubt that finding solutions to non-stationary MFGs with continuous state and action spaces is a challenging and broadly applicable problem. While the proposed method does not include any particularly novel technical ideas, it shows promising experimental results that the sample efficiency far outperforms prior methods. In particular, in a benchmark 4-room exploration MFG, the proposed DEDA-FP requires 10x less time to generate 5000 trajectories compared to standard approaches/vanilla NNs. The experiments also study a broad range of problem settings, which contain several prototypical examples of mean field games.

**Weaknesses:**

1. The proposed algorithm does not provide any theoretical guarantees, and therefore at the moment can only be classified as heuristic. Although the authors recognize this limitation eventually, there are potentially misleading statements in the paper that should be revised. In particular, below Line 146, the authors say "...the algorithm can learn the equilibrium policy...". Furthermore, in Lines 148–149, the authors say "...fully solves MFGs...". Both of these statements should be rephrased to more accurately reflect the empirical nature of the approach.

2. The authors claim that DEDA-FP can solve non-stationary MFGs. However, as far as I can tell, in all three examples, the transition kernel $P_t$ is time-dependent only through a noise term $\epsilon_t$. Admittedly, this is an important case, but the examples share essentially the same form of time dependence. Are there other examples of time dependency exhibiting a qualitatively richer or structurally different type of time variation?

3. Below Line 146, the authors claim that Algorithm 2, as opposed to Algorithm 1, can learn the equilibrium policy. It is not clear to me that Algorithm 2 has such an advantage. The authors should provide a clear explanation of why the modification introduced in Algorithm 2 is expected to yield the equilibrium policy, especially given the absence of theoretical guarantees.

4. The authors should clearly state their assumptions, for example the existence and (possibly) uniqueness of a Nash equilibrium. This is particularly relevant given the authors' use of exploitability as a metric for optimality. In particular, when should we expect a Nash policy to exist? And when do we expect such a policy to be unique?

5. Since the authors claim that a strength of DEDA-FP is its ability to handle infinite-dimensional spaces, they should state early on that policies are parameterized as neural networks. Given that policies naturally live in an infinite-dimensional space in the setting of interest, introducing this modeling choice early would help clarify the role of neural networks in the approach.

6. In the experiments, for comparison purposes I would have liked to see how Algo 3 performs compared to 1 and 2 across all runs and problem settings within the main text, if space allows. Separately, it seems that in some of the heatmap plots comparing Algo 2 and 3 (e.g. Fig 5 and 13), the equilibrium flow/equilibrium distributions obtained by Algo 3 exhibit less local density variations compared to Algo 2. Can the authors comment on this phenomenon, and is this simply a statistical artifact that arises because DEDA-FP is able to obtain many more samples than Algo 2? In particular, I would like to understand if, given the same number of samples, the equilibrium flows/distributions would look more similar between Algo 2 and 3.

Overall, the results show some promise for solving a challenging problem using deep RL, and with further examples of different time-variations and clarifications on the experimental details, I would be happy to recommend acceptance.

---

> ### Author Rebuttal · Authors · 2025-07-30
>
> We appreciate the reviewer's valuable feedback on our manuscript. We are grateful for the fact that the reviewer acknowledged our paper is *"clear in its exposition"*, that our DEDA-FP shows  *"promising experimental results that the sample efficiency far outperforms prior methods"*.
> Below, we address the weaknesses and questions.
>
> **W1 (Lack of theory):** Sorry for the misleading statements. We will rephrase them to clarify that our approach is heuristic and that the theoretical analysis is left for future work. We will also include high-level explanations in line with the answer provided to **Q1 (Error analysis) of Reviewer ZLz3**. Although this is not a full proof, we hope it will help the reader get some intuition about why the algorithm is expected to converge to an approximate Nash equilibrium, and how the different sources of error contribute to the final approximation.
>
> **W2 (Different types of time dependency):**
> In the examples, non-stationarity arises from both the noise and, in particular, the **evolution of the mean field distribution**, $\mu_t$. For instance, in the Linear-Quadratic model (Sec. 4.2), the dynamics depend on the time-dependent first moment (mean) of the population distribution, $\bar\mu_t$. This time-dependence through the evolving mean field presents a significant difficulty compared to papers focusing solely on stationary MFGs. The **Conditional** Normalizing Flow (CNF) component is precisely what allows us to learn and represent this time-dependent population distribution. It would also be possible to consider models in which the transitions are intrinsically time-dependent. For instance, still in the LQ model, we could let the coefficients $(A,B, \bar A)$ depend on time. We expect this to present less difficulty than the fact that the mean-field depends on time. We remain open to exploring other forms of time-dependency and would be interested in any further thoughts the reviewer may have.
>
> **W3 (NE policy learning):**
> Intuitively, the equilibrium is (approximately) learnt by the *average policy* $\bar\pi^K = \frac{1}{K}\sum_{k=1}^K \pi^*_k$ computed by $K$ iterations of Fictitious Play (FP). This has been proved in the finite-space case (see e.g. [Elie et al., 2020]) and for our algorithm, although we lack a full analysis, we expect a similar analysis to be possible; see again the answer to *Q1 (Error analysis) of Reviewer ZLz3*.
>
> Now the question is: *why does Algo. 2 indeed (approximately) compute the average policy, as required in FP?*
>
> The main idea is that sampling an action from $\bar\pi^K(\cdot|x)$ is equivalent to first sampling an iteration number $k \in \{1, \dots,K\}$ and then sampling an action according to the best response computed in iteration $k$, namely $\pi^*_k(\cdot|x)$. This is what is done in Algo. 1 but this requires keeping in memory all the past policies, which is inefficient.
>
> Instead, Algo. 2 does the following: at iteration $k$, it stores a batch of state-action pairs generated with $\pi ^*_k$ and keeps them in a replay buffer. Then, a policy is trained to learn the distribution of all the state-action pairs in the replay buffer.
>
> Since there is an equal probability to pick a pair generated by $\pi^*_k$ for any $k\le K$, the policy trained in this way really produces the same distribution as the theoretical average policy $\bar\pi^K$.
> Technically, Algo. 2 achieves this through the Negative Log-Likelihood (NLL) loss. For more details, we refer to line 794 of the Appendix, where we present the pseudocodes of the two algorithms.
>
> **W4 (Assumptions):**
> Thank you for raising this point. Here is a more precise and strongly justified response (that we are going to add in Section 2) regarding our assumptions, based on our discussion and the provided literature.
>
> - **Existence**: Our work assumes the existence of a Mean Field Nash Equilibrium (MFNE). This assumption is grounded in established results for related problems, as existence proofs typically rely on fixed-point arguments under specific regularity conditions on the coefficients not yet proven for our exact setting. Specifically, our justification relies on the work of [Saldi et al., 2020], who established the existence of an equilibrium for a broad class of discrete-time MFGs in an infinite-horizon, risk-sensitive setting. Crucially, their work also provides an argument that this infinite-horizon, risk-sensitive cost can be approximated by a finite-horizon (risk-neutral) game. Since our finite-horizon setting is a well-posed approximation of a problem where existence is proven, it is standard to assume that an equilibrium also exists in our case.
> Furthermore, we expect that *existence holds under mild assumptions* (typically: continuity) using the Schauder fixed point in a suitable functional space. Under more restrictive assumptions (typically: Lipschitz continuity with a small Lipschitz constant), the Banach fixed point yields both existence and uniqueness. *While a full proof is beyond the scope of the present paper, which focuses on a DRL algorithm, we included this discussion in **Section 2** of the paper, after the definition of MFNE (Definition 1)*.
>
> - **Uniqueness of Equilibrium:** For our theoretical convergence discussion, we also assume uniqueness. This is not required for the DEDA-FP algorithm itself to run. Following the approach of [Elie et al., 2020], uniqueness can be ensured by adopting the standard Lasry-Lions monotonicity assumption. In simpler terms, wherever a distribution $\mu_t$ places more agents compared to another distribution $\nu_t$, the cost to be there under $\mu_t$ must also be higher.
>
> *Please let us know if any further technical details or the precise *Assumptions* paragraph would be helpful. We would be glad to discuss this topic further*
>
> **W5 (Infinite dimensional spaces):**
> Thank you for this suggestion. We added a sentence in the abstract and at the beginning of the introduction to stress that, since the spaces are continuous, the class of policies is infinite-dimensional and hence using function approximators for the policy is unavoidable. Likewise, since we aim at learning a model for the mean-field, which is an element of $\mathcal{P}(\mathcal{X})^{N_T+1}$, using function approximators is necessary. Neural networks are natural choices due to their empirical success in a wide range of machine learning tasks.
>
> **W6 (Algo. 3 comparison and figures):**
> We thank the reviewer for their insightful comments and questions regarding the experimental comparison. We will address the two points separately.
>
> 1. **Comparison:** The full comparison results for Algo. 3 on the simpler problems are available in **Appendix C** (supp. material), but we moved them to the main text in the revision.
>
> 2. **Sampling Efficiency:** As the reviewer correctly hypothesized, plots are smoother because our method's efficiency allows for more samples within a fixed *time budget*, which we argue is the most practical basis for comparison. As shown in the table at line 266, DEDA-FP generates trajectories over 10x more efficiently. This allows it to produce a much higher-fidelity environment approximation in the same amount of time, a *core advantage* of our approach.
> To directly address the reviewer's curiosity, we produced a new plot using an equal number of samples, and the resulting distributions are indeed visually more similar. However, we believe the comparison based on computational time remains the most relevant testament to our method's advantage.
>
> **We hope that our detailed response regarding time-dependence and experimental details has fully addressed the reviewer's concerns. We hope the reviewer will consider raising their score. We look forward to any further discussion.**
>
> *References:*
>
> [Elie et al., 2020] Elie, Romuald, et al. "On the convergence of model-free learning in mean field games." Proceedings of the AAAI Conference on Artificial Intelligence. Vol. 34. No. 05. 2020
>
> [Saldi et al., 2020] Saldi, Naci, Tamer Başar, and Maxim Raginsky. "Approximate Markov-Nash equilibria for discrete-time risk-sensitive mean-field games." Mathematics of Operations Research 45.4 (2020): 1596-1620

---

> > ### Comment · Reviewer_51K9 · 2025-08-01
> > **Response to Author Rebuttal**
> >
> > Thank you for your detailed responses, it has clarified some doubts on my end. With respect to existence and uniqueness of the Nash policy, the ability of Alg 2 to learn the policy and the experimental details, my doubts have been fully clarified. I think the discussion with Rev. Zlz3 would be useful to add to the manuscript to give some directions for a potential theoretical convergence guarantee. That being said, I still have some questions about the time-dependency of the system, as follows:
> >
> > 1. Would the convergence be sharper (or maybe even the theoretical analysis simpler) if the time dependence on the mean field is structured in a predictable/learnable way? For instance, if it is periodically changing over time or near-periodic?
> > 2. One of the other structures I had in mind was if only some or one of the parameters was time-dependent (as you have mentioned in your response), while the mean-field was stationary. I would be curious to see if this would result in better empirical performance for DEDA-FP?
> >
> > Thank you!

---

> > > ### Author Response · Authors · 2025-08-01
> > > **Further clarifications on time-dependency**
> > >
> > > Thank you for your feedback and the new questions. We reply below to each question.
> > >
> > > **Q1**. You are right to suggest that exploiting specific structures in the time-dependency of the mean field can lead to simpler analysis and improved empirical performance. We can elaborate different scenarios:
> > > 1. **Turnpike Phenomenon:** In certain long-horizon MFGs, the system can exhibit a turnpike phenomenon, where the mean field evolves significantly near the start and end of the horizon but remains quasi-stationary in between. If this was the case, the learning process could be made more efficient. For instance, instead of a uniform time discretization, we could use a non-uniform grid for our neural networks: using more network parameters to capture the rapid changes at the beginning and end, and fewer for the stable mid-horizon phase. This would improve efficiency in both memory and computation. This phenomenon has been studied in works such as [Trusov, 2020] and [Cirant et al., 2021].
> > >
> > > 2. **Periodic Evolution:** You mentioned the interesting case where the mean field evolution is periodic. If the period were known, one could simply solve the MFG over a single period and replicate the solution to understand the long-term behavior. If the period were unknown, one could imagine an algorithm that first learns this periodicity before applying the single-period solver. We have not investigated this direction for now but we are aware of some works on MFG with periodicity in time, such as [Cirant et al., 2018].
> > >
> > > **Q2.** If the mean field was stationary, one could simply train a *single* Normalizing Flow (without conditioning) and then use it at every time step. This would be more efficient in terms of memory and computational time than our current implementation. However, to the best of our knowledge, MFG with stationary mean field were always studied when the coefficients themselves were stationary (i.e., time-independent) too; see for example [Guo et al, 2019],  [Subramanian et al., 2019] and [Perrin et al., 2020, appendix]. We do not know any setting where the coefficients are time-dependent but the mean field is not. One idea, following [Perrin et al., 2020; Appendix C], could be to consider an infinite horizon discounted MFG with time-dependent coefficients and still define the $\gamma$-discounted distribution, that we can use in the transitions and rewards. Then the players perceive only a stationary mean field. If the reviewer believes this example would add value to the paper's contribution, we would be happy to formulate and include it in the final manuscript (but we will not be able to show new plots during the rebuttal, as per the conference's policy).
> > >
> > > We hope these clarifications regarding the time-dependent nature of the mean field game problem have fully addressed your curiosity and we hope you consider to raise your score. We welcome the opportunity to discuss these points further.
> > >
> > > *References*
> > >
> > > [Cirant et al., 2021] Cirant, Marco, and Alessio Porretta. "Long time behavior and turnpike solutions in mildly non-monotone mean field games." ESAIM: Control, Optimisation and Calculus of Variations 27 (2021): 86.
> > >
> > > [Trusov, 2020] Trusov, N. V. "Numerical solution of Mean Field Games problems with turnpike effect." Lobachevskii Journal of Mathematics 41.4 (2020): 561-576
> > >
> > > [Cirant et al., 2018] Cirant, Marco, and Levon Nurbekyan. "The variational structure and time-periodic solutions for mean-field games systems." MINIMAX THEORY AND ITS APPLICATIONS 3.2 (2018): 227-260.
> > >
> > > [Guo et al, 2019] Guo, Xin, et al. "Learning mean-field games." Advances in neural information processing systems 32 (2019)
> > >
> > > [Subramanian et al., 2019] Subramanian, Jayakumar, and Aditya Mahajan. "Reinforcement learning in stationary mean-field games." Proceedings of the 18th international conference on autonomous agents and multiagent systems. 2019

---

> ### Comment · Reviewer_51K9 · 2025-08-05
> **Response to follow-ups**
>
> Thanks for your response! I think it is valuable to have a deeper conversation about the nature of nonstationarity in MFGs, and the examples you have provided are definitely interesting for future consideration. That being said, the convergence guarantee you have given in the latest response to Reviewer Zlz3 is definitely more crucial to be added to the present draft. The proof idea seems correct, and relies on similar techniques in the literature (e.g. Elie et al 2020). I do have one small question -- are there any known results for how to estimate or control the distribution error induced by the CNF? This could motivate the study of how to appropriately bound each of the error terms in a statistical learning sense, which has been explored before in the RL literature (e.g. [1]). This general line of inquiry is, of course, very challenging from a theoretical perspective but certainly something to consider for future work.
>
> Overall, adding this to the paper would improve the quality of the submission and give credence to the experimental results. For this reason I am happy to increase my score to an accept.
>
> [1] Farahmand, A. M., Szepesvári, C., & Munos, R. (2010). Error propagation for approximate policy and value iteration. Advances in neural information processing systems, 23.

---

> > ### Author Response · Authors · 2025-08-06
> >
> > We sincerely thank the reviewer for raising their score and for the interesting research direction about controlling the distribution error from the Conditional Normalizing Flow (CNF), $\epsilon^k_{cnf}$.
> >
> > The specific architecture we use is a time-conditioned Normalizing Flow, which is essential for capturing the non-stationary dynamics of the MFG. To the best of our knowledge, a formal statistical analysis of the convergence rates for such time-conditioned architectures is still an open and exciting research area.
> >
> > In this direction we can draw some parallels with the recent theoretical advances in standard (non-conditioned) normalizing flow. In particular, the specific architecture we use, Neural Spline Flows, has fundamental representation capabilities. For instance, as [Durkan et al., 2019] argue, a differentiable monotonic spline with sufficiently many bins can approximate any differentiable monotonic function in a bounded interval, and this can be used to approximate density functions.
> >
> > However, as the reviewer mentioned, a complete analysis would require extending these results to time-conditioned flows and then propagating each of the three error terms ($\epsilon_{br}^k$, $\epsilon_{sl}^k$, $\epsilon_{cnf}^k$) through the steps of the algorithm (in the spirit of [Farahmand et al., 2010]) to provide an overall sample complexity for achieving an $\epsilon$-Nash equilibrium. We agree that this is an important direction for future work and we will mention this in the revised paper.
> >
> > *References:*
> >
> >
> > [Durkan et al., 2019] Durkan, C., Bekasov, A., Murray, I., & Papamakarios, G. (2019). Neural spline flows. Advances in neural information processing systems, 32.
> >
> > [Farahmand et al., 2010] Farahmand, A. M., Szepesvári, C., & Munos, R. (2010). Error propagation for approximate policy and value iteration. Advances in neural information processing systems, 23.

---

### Note · Authors · 2025-08-12

We sincerely thank all the reviewers for their time and for providing detailed, constructive comments that have helped significantly improve our paper.

The discussion period was highly productive. In particular, have:

- Outlined a **proof of convergence** for DEDA-FP, extending the framework of Elie et al. (2020) to our setting; this addresses the reviewers' primary concern regarding the lack of theoretical guarantees.

 - Clarified the **design choices** of DEDA-FP and the logic behind our average policy learning scheme.

- Clarified the **comparison with the literature**, in particular the relevance of our numerical examples compared with the state-of-the-art in this field.


We will include the changes we mentioned during the discussion in the final paper.

We are confident that the strengthened theoretical guarantees, clarified methodological choices, and expanded literature comparisons address the main concerns raised. These additions complement our original contributions: a scalable algorithm for **non-stationary continuous-space** mean field games that combines deep reinforcement learning, supervised learning for policy averaging, and density modeling to achieve accurate population dynamics, **improved scalability, and faster sampling**. The new **convergence (error-propagation) analysis**, which will be included in the final version, further solidify the paper’s theoretical and practical contributions.

We are very grateful that two reviewers (**51K9** and **ZLz3**) acknowledged that these additions significantly strengthen the paper and have raised their scores to "**Accept**." We are hopeful that our comprehensive responses, particularly the new convergence analysis, will convince reviewer **uqaE** and the Area Chair of our paper's merits.

---

### Decision · Program_Chairs · 2025-09-17

**Decision:**

Accept (poster)

**Comment:**

The paper makes progress by developing a novel deep RL algorithm specifically designed for non-stationary continuous MFGs. After a careful rebuttal process a clear consensus emerged amongst the reviewers that the contributions of the paper were significant enough to merit acceptance.